# The role of metabolism in shaping enzyme structures over 400 million years

Oliver Lemke[1,2,15], Benjamin Murray Heineike[1,3,4,15], Sandra Viknander[5], Nir Cohen[1], Feiran Li[5], Jacob Lucas Steenwyk[6,7,8,9], Leonard Spranger[1], Federica Agostini[1], Cory Thomas Lee[1], Simran Kaur Aulakh[3,4], Judith Berman[10], Antonis Rokas[7,9], Jens Nielsen[5], Toni Ingolf Gossmann[11], Aleksej Zelezniak[5,12,13] & Markus Ralser[1,2,3,4,14 ✉]

Advances in deep learning and AlphaFold2 have enabled the large-scale prediction of protein structures across species, opening avenues for studying protein function and evolution[1]. Here we analyse 11,269 predicted and experimentally determined enzyme structures that catalyse 361 metabolic reactions across 225 pathways to investigate metabolic evolution over 400 million years in the Saccharomycotina subphylum[2]. By linking sequence divergence in structurally conserved regions to a variety of metabolic properties of the enzymes, we reveal that metabolism shapes structural evolution across multiple scales, from species-wide metabolic specialization to network organization and the molecular properties of the enzymes. Although positively selected residues are distributed across various structural elements, enzyme evolution is constrained by reaction mechanisms, interactions with metal ions and inhibitors, metabolic flux variability and biosynthetic cost. Our findings uncover hierarchical patterns of structural evolution, in which structural context dictates amino acid substitution rates, with surface residues evolving most rapidly and small-molecule-binding sites evolving under selective constraints without cost optimization. By integrating structural biology with evolutionary genomics, we establish a model in which enzyme evolution is intrinsically governed by catalytic function and shaped by metabolic niche, network architecture, cost and molecular interactions.

Enzymes evolve as part of the metabolic network, a large, interconnected system that possesses a topology dependent on evolution and the chemical properties of its metabolites[3–5]. Because of the central role of metabolism, enzymes are important drug targets, biomarkers and a focus of bioengineering[6–8]. Despite its critical role across disciplines, our understanding of the global biochemical constraints that shape enzyme function—and therefore its evolution—remains incomplete.

Comparing enzyme sequences across evolution has revealed various constraints that act at the amino acid level, such as the chemical identity of side chains and epistatic interactions[9–11]. Moreover, the sequence of enzymes is shaped by the costs of enzyme production[12]. Indeed, metabolic cost optimization is observed at the species and molecular levels. For example, under many conditions, cells prefer cost-effective fermentation over oxidative metabolism, despite the latter producing stoichiometrically higher ATP amounts[13]. Furthermore, especially high-abundance enzymes have evolved by incorporating less energetically costly amino acids[14–16].

We hypothesized that the systematic accessibility of protein structures enabled by structural prediction[1] would allow us to integrate structural biology with evolutionary genomics and expand our understanding of the relationship between metabolism and protein evolution. We leveraged the extensive characterization of the Saccharomycotina subphylum, which represents 400 million years of evolution and includes *Saccharomyces cerevisiae* and *Candida albicans*[2,17–19], and examined 11,269 AlphaFold2-predicted and experimentally determined enzyme structures that belong to 424 orthologue groups (orthogroups) associated with 361 metabolic reactions in 224 metabolic pathways. Linking these structures with phenotypic data, enzyme properties and metabolic network reconstructions, we identified structural changes that are associated with metabolic constraints. We report how structural evolution depends on metabolic properties across species, pathways and molecular levels.

[1]Department of Biochemistry, Charité-Universitätsmedizin Berlin, Berlin, Germany. [2]Exploratory Diagnostic Sciences, Berlin Institute of Health at Charité, Berlin, Germany. [3]Center for Human Genetics, Nuffield Department of Medicine, University of Oxford, Oxford, UK. [4]Molecular Biology of Metabolism Laboratory, The Francis Crick Institute, London, UK. [5]Department of Life Sciences, Chalmers University of Technology, Gothenburg, Sweden. [6]Howard Hughes Medical Institute, University of California Berkeley, Berkeley, CA, USA. [7]Department of Biological Sciences, Vanderbilt University, Nashville, TN, USA. [8]Department of Molecular and Cell Biology, University of California Berkeley, Berkeley, CA, USA. [9]Evolutionary Studies Initiative, Vanderbilt University, Nashville, TN, USA. [10]Shmunis School of Biomedical and Cancer Research, George S. Wise Faculty of Life Sciences, Tel Aviv University, Ramat Aviv, Israel. [11]Faculty of Biochemical and Chemical Engineering, TU Dortmund University, Dortmund, Germany. [12]Institute of Biotechnology, Life Sciences Centre, Vilnius University, Vilnius, Lithuania. [13]Randall Centre for Cell & Molecular Biophysics, King's College London, London, UK. [14]Max Planck Institute for Molecular Genetics, Berlin, Germany. [15]These authors contributed equally: Oliver Lemke, Benjamin Murray Heineike. ✉e-mail: markus.ralser@charite.de

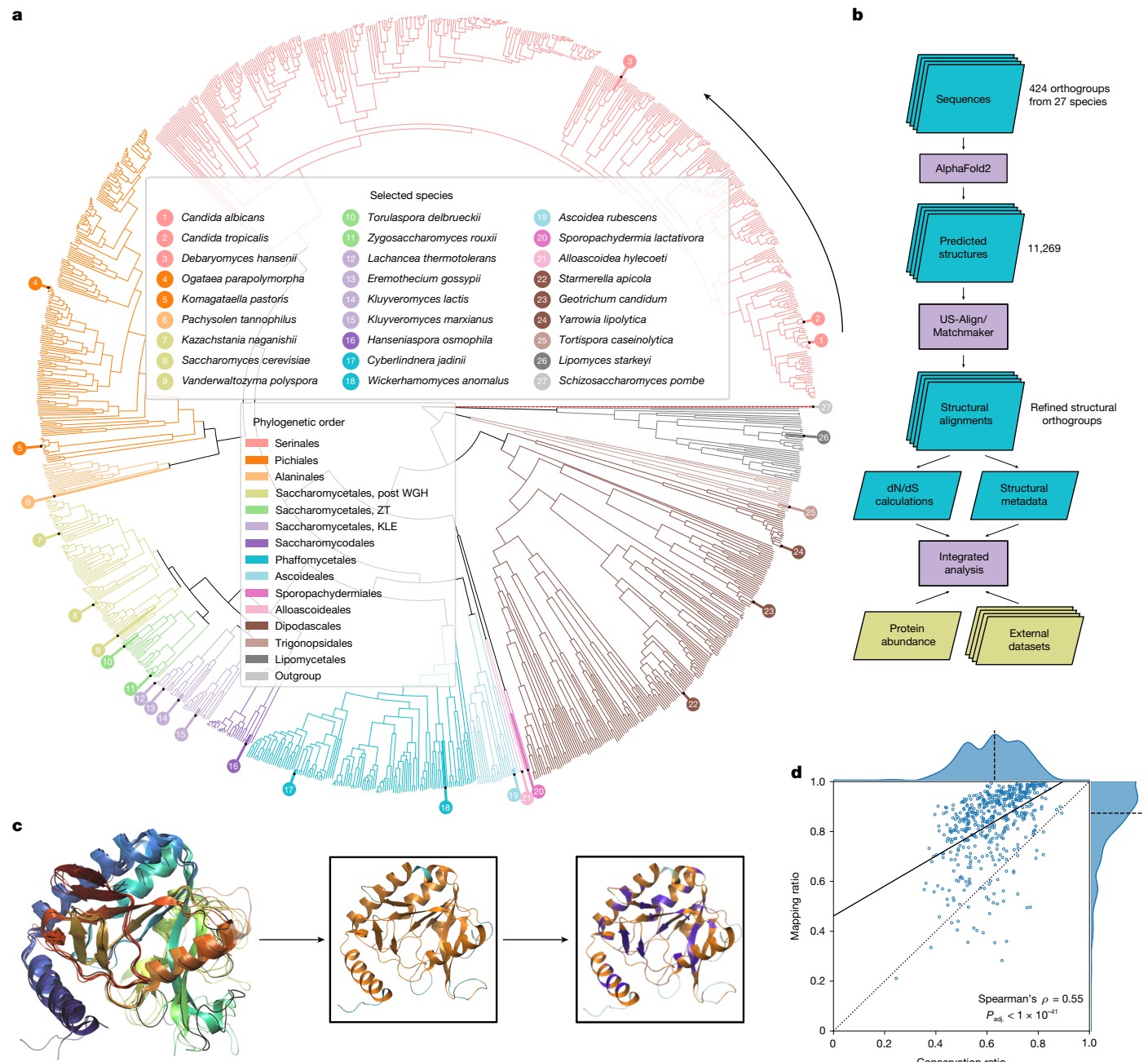

**Fig. 1 | Divergence in structurally conserved regions corresponds to metabolic properties acting at the species, pathway and molecular levels.**
**a**, Phylogenetic tree of the Saccharomycotina yeast subphylum highlighting the 26 species (+1 outgroup species) for which metabolic enzyme structures were generated and systematically compared. Colours indicate phylogenetic order, numbering counterclockwise starting at *C. albicans*. Branch lengths and topology are from the species time tree as calculated in ref. 2, except for the branch for the outgroup species *S. pombe*, which is not drawn to scale. **b**, Illustration of our analysis pipeline. **c**, Example alignment for 5-formyltetrahydrofolate cyclo-ligase structures in five species (*S. cerevisiae*, *C. albicans*, *Kluyveromyces lactis*, *Kluyveromyces pastoris* and *S. pombe*). The black line denotes the reference structure in *S. cerevisiae* (Fau1p). Insets show the orthologue from *C. albicans* with mapped residues (*M*, orange) and unmapped residues (*N*, cyan), as well as residues conserved (*C*, purple) between *S. cerevisiae* and *C. albicans* (purple). **d**, Mean mapping ratio ($M/(M + N)$) to mean conservation ratio ($C/M$) for the 529 reference structures that passed our filters. Dotted line denotes the identity line, the dashed line denotes the axis median and the solid line indicates the best linear fit. WGH, whole-genome hybridization.

## Mapping evolution in enzyme structures

We selected 26 out of 332 highly phylogenetically diverse yeast species of the Saccharomycotina subphylum[19,20] (Supplementary Note 1) and included the model fission yeast *Schizosaccharomyces pombe* as an outgroup to root protein trees (Fig. 1a and Supplementary Table 1.1). Then, for enzymes present in the YeastPathways database, we initially assigned orthologues on the basis of sequence-based clusters[19].

From these sequences, we obtained 1,301 structures from Alpha-FoldDB[21] and a further $n$ = 9,968 structures that we predicted using AlphaFold v.2.0.1 at the start of our project (Supplementary Note 2). Our final dataset consisted of 11,269 enzyme structures organized in 424 orthogroups.

Prediction quality was assessed using the predicted local distance difference test (pLDDT) score[22,23], which revealed that our dataset included well-structured proteins (mean pLDDT = 90.4, mean coefficient of

variation (c.v.) = 0.15). Compared with the overall structures, the terminal regions were predicted with a lower quality (first 10% of the sequence: mean pLDDT = 79.1, mean c.v. = 0.29; central 80%: mean pLDDT = 92.2, mean c.v. = 0.13; last 10%: mean pLDDT = 87.3, mean c.v. = 0.18) (Extended Data Fig. 1a). Then, we validated and refined orthogroup assignments using hierarchical clustering on the basis of a bidirectional template modelling score[24] (Extended Data Fig. 1b,c). On the basis of a linkage cut-off of 0.2, 29 sequence-based orthogroups were split into distinct structural orthogroups (Supplementary Table 1.5), which improved the average template modelling score (from 0.71 to 0.77).

To benefit from the extensive characterization of *S. cerevisiae*, we calculated pairwise alignments for each orthogroup to *S. cerevisiae* enzyme structures using the matchmaker algorithm of UCSF Chimera[25]. To link these structures to metabolic constraints (Fig. 1b), we calculated averaged mapping ratios (MRs) and conservation ratios (CRs). The MR quantifies the percentage of amino acids that are 1:1 mappable to a *S. cerevisiae* enzyme structure, whereas the CR quantifies the percentage of mapped residues identical to those of the reference structure. The CR was tightly correlated with a CR based on the amino acid types (Extended Data Fig. 1d), thus capturing physicochemical properties. In agreement with this, a strong inverse relationship to the change of the octanol–water partition coefficient is observed (Extended Data Fig. 1e; $P < 1 \times 10^{-31}$, Spearman's $\rho = -0.49$).

We illustrate the MR and CR for the orthogroup of 5-formyltetrahydrofolate cyclo-ligase (27 structures, CR = 0.40, MR = 0.87; Fig. 1c and Extended Data Fig. 1f; Fau1p in *S. cerevisiae*). The core structures of the enzymes map well and the secondary structural elements (helical or extended) revealed a mean MR of 95.4%, whereas the regions without secondary structures (random coils), which have a higher conformational flexibility, have a mean MR of 77.3%. The median MR for all orthogroups was 87.4% (interquartile range (IQR) = 78.3–93.9%) (Fig. 1d), with missing mapping mainly in low-pLDDT scoring regions (Extended Data Fig. 1g), that is, the carboxy and amino termini (Extended Data Fig. 1a) and random coil regions (60% of the unmapped and 36% of the mapped regions). Although the MR generally correlated with the CR (Spearman's $\rho = 0.55$, adjusted $P (P_{adj.}) < 1 \times 10^{-41}$) (Fig. 1d), both the MR and the CR reflect different properties of structural divergence. As the larger structural rearrangements reflected by the MR were less frequent in our orthogroups, we focused on the CR, for which we observed a high degree of diversity (median CR = 62.9%, IQR = 53.6%, 71.2%, total range = 24.5%, 89.2%) that could be linked to metabolic evolution. Here we refer to sequence divergence in structurally mapped regions as divergence (low CR) and sequence similarity in structurally mapped regions as conservation (high CR).

## Biochemical constraints

### The impact of metabolic specialization

We asked whether metabolic specializations at the species level are reflected in the protein structures and linked divergence to the growth properties of yeast in 21 different carbon sources[17–19,26] (Extended Data Fig. 2a). Enzymes of species able to ferment glucose, raffinose, galactose and sucrose exhibited the smallest *P* values for differences in average CR between subgroups, alongside enzymes from species that grew aerobically on D-xylose ($P_{adj.} < 1 \times 10^{-83}$, two-sided Wilcoxon signed-rank test). Enzymes from anaerobically fermenting species had a higher conservation relative to the structure from *S. cerevisiae*, which also ferments (Fig. 2a and Extended Data Fig. 2b). Although this finding corresponds to their closer phylogenetic relationship, some of the largest differences in CR were detected in the orthogroups of enzymes involved in central carbon metabolism and the electron transport chain (ETC), for example, Kgd2p (tricarboxylic acid (TCA) cycle) and Cox7p (respiratory chain). These more divergent orthogroups also included Met10p (methionine and sulfur cycle), Ath1p (trehalose

metabolism) and Erg1p (ergosterol biosynthesis). We also observed cases in which the CR in non-fermenting species was higher than in the fermenting species. Again, the enzymes were directly related to oxidative metabolism, including Ndi1p, Ald5p, Idp1p and Ilv6p. Moreover, gene ontology (GO)-slim terms 'membrane', 'lipid metabolism', 'endoplasmic reticulum' and 'endomembrane system' were enriched in the first quartile of orthogroups with the largest differences in CR between subgroups for glucose fermentation ($P_{adj.} < 1 \times 10^{-2}$, Fisher's exact test) (Extended Data Fig. 2c).

To study an example of metabolic specialization, we focused on the xylose use pathway. Of the 26 species examined, 12 can grow on D-xylose, 8 cannot and 6 have a conditional phenotype (Extended Data Fig. 2a). Several enzymes required for xylose use, such as transketolase, enzymes in the thiamine biosynthetic pathway and the ETC, were among those with the highest change in CR (Fig. 2b). Notably, the CR measured in relation to the two *S. cerevisiae* acetyl-CoA synthase paralogues, the aerobic (Acs1p) or the anaerobic (Acs2p), behaved differently depending on the capacity to use xylose (Fig. 2b, Extended Data Fig. 2d,e and Supplementary Note 3.1), indicating specialization for Acs1p. Thus, species that specialize metabolically show different patterns of divergence in enzymes that are related to the relevant metabolic traits.

### The impact of pathway membership

Next, we projected diversity in the orthogroups onto a genome-scale reconstruction of the metabolic network (Fig. 2c). Then, we performed a pathway enrichment analysis on the 25% most divergent and conserved enzymes (Extended Data Fig. 3a,b). The most conserved enzyme structures belonged to pathways for purine biosynthesis, specific amino acid biosynthesis as well as central metabolism ($P_{adj.} < 0.05$, Fisher's exact test; Extended Data Fig. 3a,b). The same pathways also showed early enrichment in the receiver operating characteristic curve and high area under the curve (AUC) values (Fig. 2d and Extended Data Fig. 3c). Consistently, these orthogroups were enriched in the GO-slim terms 'generation of precursor metabolites and energy' and 'nucleobase-containing small molecule metabolic process'. The most divergent enzymes tended to be more broadly distributed across the metabolic network, with the exception that enzymes belonging to 'lipid metabolic process' were enriched. For instance, Oar1p (CR = 0.384), Tes1p (CR = 0.391) and Eci1p (CR = 0.441) for fatty acid oxidation and Tsc10p (CR = 0.411) for sphingolipid synthesis were within the top-5%-quantile of divergence. We speculate that lipidomes of cells have a greater flexibility to adapt to the metabolic environment. Furthermore, the UDP-*N*-acetylglucosamine transferases Alg13p and Alg14p, related to N-linked glycosylation, were also among the most divergent orthogroups.

### The impact of molecular function

Next, we annotated the biochemical activities by extracting enzyme classification (EC) identifiers from UniProtDB. Of 468 matching EC entries, 238 (51%) were supported by a direct PubMed Evidence code. Our enzyme structures with functional annotation covered 44% (361 out of 817) of all EC classifiers, distributed across 224 metabolic pathways (*S. cerevisiae* annotation; Supplementary Table 1.4) and encompassed all major enzyme classes, including 119 oxidoreductases, 191 transferases, 55 hydrolases, 49 lyases, 21 isomerases and 29 ligases. The dataset also contains two translocases, but, because of their low number, we refrained from making general conclusions for this EC class.

We noted that in each orthogroup, most enzymes seem to catalyse the same or highly similar reactions, and observed no changes in EC function at levels 1, 2 or 3. A change in the EC level 4 classification was detected in only five orthogroups. For example, in OG1390, containing Hsu1p, Str2p and YML082p in *S. cerevisiae*, a slight change in the catalysed reaction occurs[27,28], which might be facilitated by changes in a domain close to the binding site (Supplementary Note 3.2).

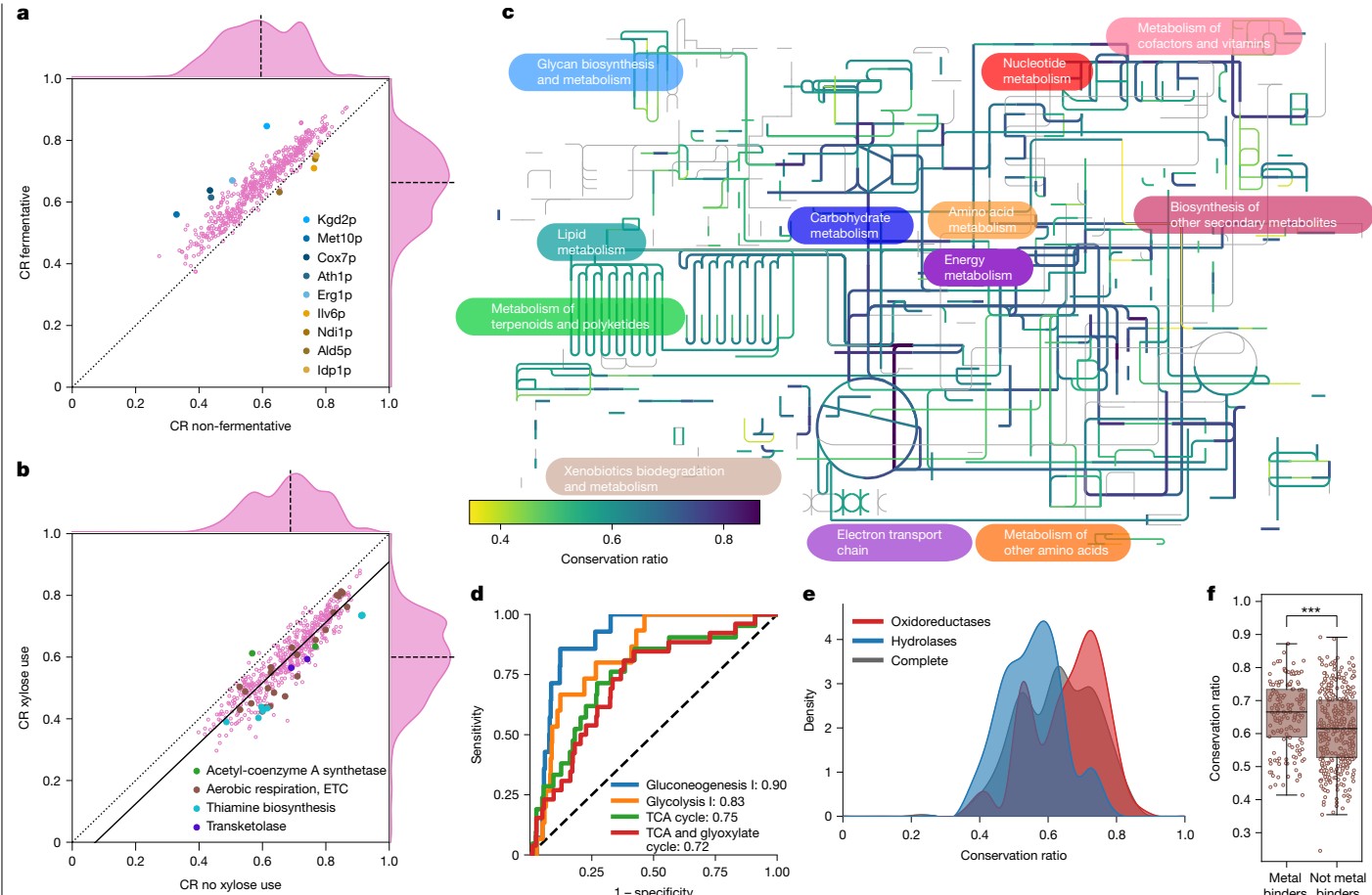

**Fig. 2 | Metabolic network organization constrains the structural evolution of enzymes in Saccharomycotina. a,b,** Mean conservation ratio per protein calculated for species that do or do not ferment glucose (**a**) and species that can or cannot grow on D-xylose (**b**), in refs. 17,18. For this analysis, orthogroups were temporarily subdivided on the basis of the phenotype of the species. For **a**, proteins with the largest differences in both directions, and, for **b**, pathways with remarkable changes, are highlighted. The dotted line denotes the identity line, the solid line denotes the linear fit and the dashed line denotes the axis median. **c,** Conservation ratio projected onto the yeast metabolic map of iPath3. **d,** Receiver operating characteristic (ROC) curve for four highly enriched pathways covering the TCA cycle and glucose metabolism. The number in the legend indicates the AUC. The dashed line denotes the identity line representing random sampling. **e,** Distribution of the mean conservation ratio of orthogroups assigned to oxidoreductases (red) or hydrolases (blue). The average distribution for all orthogroups is shown in black. **f,** Conservation ratio of enzymes known to bind metals ($n = 158$) and those not known to bind metals ($n = 371$), expressed as a box plot. The line denotes the median and the boxes denote the first and third quartile, the whiskers extend up to 1.5 times the IQR. Each dot represents an orthogroup. ***$P < 1 \times 10^{-4}$, two-sided Wilcoxon–Mann–Whitney $U$-test.

We detected a clear relationship between enzyme class and diversity. Oxidoreductases are enriched for high CR, and hydrolases for low CR (Fig. 2e and Extended Data Fig. 3b). The high conservation of oxidoreductases was explained by their prominent role in glycolysis, gluconeogenesis and the TCA cycle. After excluding these central metabolic pathways, oxidoreductases were not more conserved than other enzymes (Extended Data Fig. 3d). Conversely, the increased diversity of hydrolases was not explained by their role in specific metabolic pathways (Extended Data Fig. 3e).

Furthermore, we report a role for non-catalytic protein–small molecule interactions. First, metal-binding enzymes are more conserved than non-metal-dependent enzymes (two-sided Wilcoxon–Mann–Whitney $U$-test, $P < 1 \times 10^{-4}$, 7.6% decrease in median conservation, Cliff's $\Delta = 0.23$; Fig. 2f). Furthermore, we observed that enzymes with a higher number of intracellular inhibitors[29] are more conserved (Kendall $\tau = 0.22$, $P_{adj.} < 1 \times 10^{-4}$; Extended Data Fig. 3f). For instance, Gnd2p, a central enzyme of the pentose phosphate pathway, is inhibited by at least 45 cellular metabolites[29], and its orthogroup was one of the least divergent. We thus concluded that characteristics of enzymes, such as pathway membership, dependency on metal ions and the number of small molecule interactions, constrain divergence.

## Abundance and flux diversity constraints

Protein abundance is important for sequence conservation[30,31]. Given that enzyme expression is contingent on the specific activity and flux of the enzyme, we hypothesized that abundance could be a mechanism through which metabolism influences structural evolution. We obtained nine of the examined species from the UK National Collection of Yeast Cultures[20] (Fig. 3a) and used proteomics to estimate their protein abundance (Fig. 3a, Extended Data Fig. 4a,b and Supplementary Methods). We found that low-abundance enzymes were more diverse than high-abundance enzymes (Fig. 3b; Spearman's $\rho = 0.48$, $P_{adj.} < 1 \times 10^{-27}$). This relationship was dependent on the enzyme class; abundance and diversity were highly interdependent for isomerases, but not for hydrolases (Fig. 3c).

Next, we estimated metabolic flux through each pathway on the basis of genome-scale metabolic models for each of the 26 species[18]. We benchmarked the fluxes to $^{13}$C flux measurements for central metabolites, which are captured by both approaches, and obtained good agreement (Pearson's $r = 0.8$ for median ($^{13}$C flux); Extended Data Fig. 4c). Notably, we revealed only a weak correlation between flux and CR (Fig. 3d; Spearman's $\rho = 0.19$, $P_{adj.} < 1 \times 10^{-4}$). Instead, a stronger signal

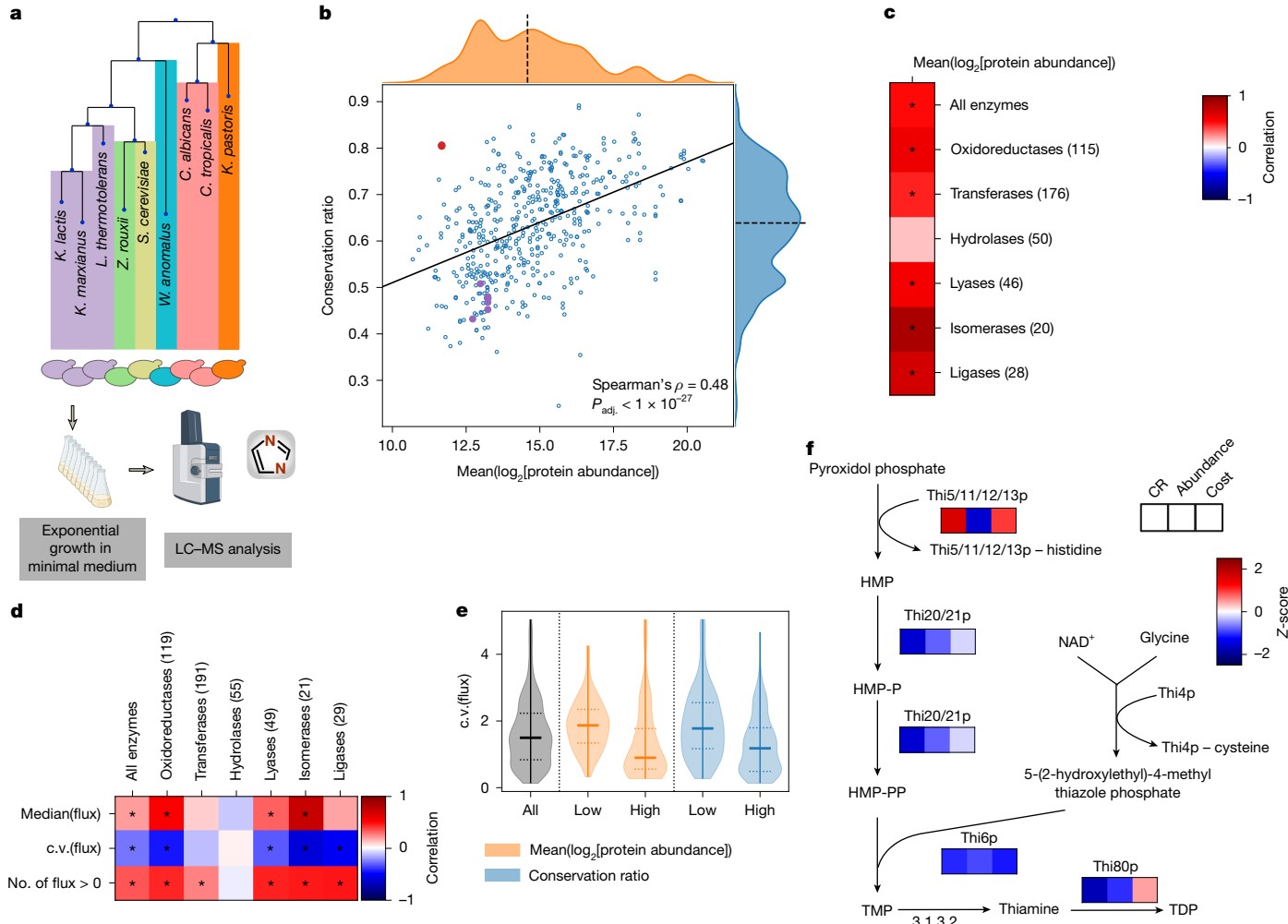

**Fig. 3 | Roles of enzyme abundance and flux in structural evolution.**
**a**, Protein abundance was determined using proteomics with data-independent acquisition (Supplementary Methods). **b**, Mean conservation ratio and the mean $\log_2$-transformed protein abundances for 9 of the 27 investigated species measured during exponential growth in minimal medium ($n = 491$). The orthogroup containing the Thi5/11/12/13p family is highlighted in red, other enzymes of the thiamine biosynthetic pathway are highlighted in purple. The solid line indicates the best linear fit, the dashed line denotes the axis median. **c**, Spearman's correlation between protein abundance and conservation ratio for all tested enzymes as well as broken down according to enzyme class (numbers are adjusted according to the presence of protein abundance data). *$P_{adj.} < 0.05$. **d**, Correlation between different measures of predicted flux (median, c.v. and number of species ($n = 329$) with flux through a given orthogroup) and conservation ratio for all tested enzymes, broken down in each column according to the enzyme class. For the flux median and c.v., the Spearman's

correlation was used, and the Kendall $\tau$ correlation was used for the number of species. The number in brackets indicates the number of enzymes per class. *$P_{adj.} < 0.05$. **e**, Violin plot of the c.v. of the fluxes for the orthogroups in the first and last quartile of conservation ratio (blue) or protein abundance (orange). **f**, The thiamine biosynthesis pathway is shown. Heat maps underneath enzymes indicate the $Z$-scores of the mean conservation ratio, mean $\log_2$-transformed protein abundance and averaged cost. The Thi5/11/12/13p family and Thi4p undergo suicide reactions in which they lose a histidine and cysteine residue, respectively. Illustrations in **a** were created using BioRender. Heineike, B. (2025) https://BioRender.com/r831qhq. HMP, 4-amino-2-methyl-5-pyrimidine; HMP-P, 4-amino-2-methyl-5-pyrimidine phosphate; HMP-PP, 4-amino-2-methyl-5-pyrimidine diphosphate; LC–MS, liquid chromatography–mass spectrometry; NAD$^+$, nicotinamide adenine dinucleotide; TDP, thiamine diphosphate; TMP, thiamine phosphate.

was detected for flux variability (Spearman's $\rho = -0.27$, $P_{adj.} < 1 \times 10^{-8}$). Consistently, orthogroups with low CR exhibited a wide range of fluxes (Fig. 3e and Extended Data Fig. 4d). There were exceptions to this trend, such as in the orthogroup containing the sphingosine kinase Ysr3p, indicating highly variable flux and high diversity (low CR), despite high abundance. Moreover, we also detected a dependence on the enzyme class. Although divergence was strongly linked to the flux carried by oxidoreductases, for which all tested measures (median flux, variability of the flux and species in which flux was present for the orthogroup) correlated with conservation, other enzyme classes, especially hydrolases (Fig. 3d; $P_{adj.} > 0.7$ for all three measures), lacked these relationships. We also estimated enzyme processivity ($k_{cat}$) for each protein sequence in a species-specific manner[32], and found a weak

relationship between conservation and $k_{cat}$. However, we report that the log-transformed variability (standard deviation) of $k_{cat}$ is higher in orthogroups that are diverse (Extended Data Fig. 4e; Spearman's $\rho = -0.27$, $P_{adj.} < 1 \times 10^{-8}$). Thus, both flux and $k_{cat}$ seem to be associated with structural evolution primarily through their variability, rather than their absolute values.

In parallel, we noticed that enzymes can escape the typical relationship between abundance and conservation due to unique functional constraints. Our attention was drawn to the thiamine (vitamin B1) biosynthetic pathway, for which the orthogroup containing Thi5p, Thi11p, Thi12p and Thi13p was highly conserved, despite low abundance (Fig. 3b). Notably, thiamine is extremely energetically costly to synthesize, as two of its reaction steps are catalysed by suicide enzymes that

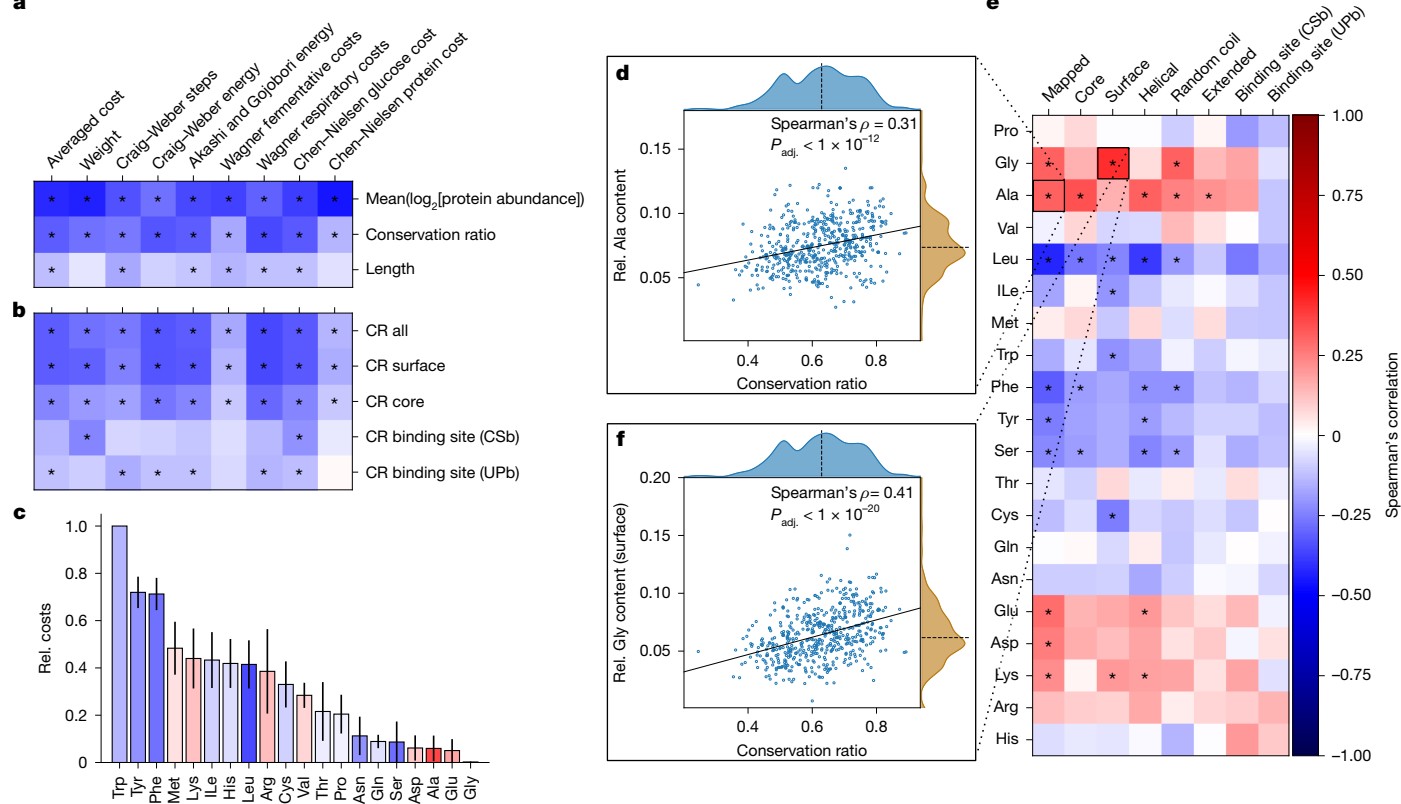

**Fig. 4 | Evolutionary cost optimization acts differently depending on the structural element and amino acid properties. a,b**, Spearman's correlation of the average cost per amino acid for the entire protein compared with the mean conservation ratio, $\log_2$-transformed protein abundance and length of the protein chain (**a**) and compared with the mean CR of selected structural features (**b**). Cost measures have been described previously[12,14,34,36]. *$P_{adj.} < 0.05$. **c**, Median normalized cost (Rel. cost) of each amino acid sorted in descending order. Error bars denote the median absolute deviation. Bar colouring indicates the Spearman's correlation between the mean CR and the relative amino acid content of the entire protein. **d–f**, Spearman's correlation between the relative amino acid content for various structural features of proteins compared with the mean overall CR (**e**), scatter plot illustrating the relationship between mean overall CR and the relative (Rel.) alanine content of the entire mapped region (**d**) and relative surface glycine content of the mapped region (**f**). Solid line indicates the best linear fit and the dashed line denotes the axis median (**d** and **f**). *$P_{adj.} < 1 \times 10^{-4}$ (**e**). The colour bar shown in **e** applies to **a–c** and **e**.

lose an essential amino acid after catalysis[33] (Fig. 3f). This orthogroup thus illustrated that evolutionary diversification is not necessarily directly constrained by abundance, but more likely by cost.

## Hierarchy of cost optimization

We thus calculated the average cost per amino acid for each protein averaged over the orthogroup using established cost metrics[12,14,15,34–36] (Supplementary Note 4). Previous studies indicated that high-abundance proteins evolve a less costly amino acid composition[14–16]. Consistently, low-abundant, diverse enzymes had more costly amino acid compositions than high-abundant enzymes (Fig. 4a). However, the overall protein chain length, a proxy for the total protein cost, showed only a weak correlation with the mean averaged cost per amino acid (Spearman's $\rho = -0.13$, $P_{adj.} < 1 \times 10^{-2}$), indicating that cost optimization might be due to selection acting on specific structural elements. Notably, these relationships were indicated by cost models that are both dependent and independent of species-specific metabolic networks (Supplementary Note 4).

To test the role of structural elements, we next identified small-molecule-binding sites using two orthogonal strategies: (1) binding-site information deposited in UniProt to account for directly coordinating residues (UPb) and (2) residues near small molecules from experimentally determined structures obtained from the RCSB Protein Data Bank (PDB) to also account for the physicochemical environment (CSb). Next, to distinguish between surface and core residues, we used the relative amino acid solvent-accessible surface area and designated each residue as either a surface-exposed or core residue[37]. We found that the amino acid composition is optimized differently in these structural elements. In general, the average cost of core residues was higher than the cost of surface residues, whereas binding sites had a more variable cost. Among surface residues, we also observed increased costs for membrane-bound proteins and short proteins (<100 amino acids), such as Qcr8p or Kti11p in *S. cerevisiae*, that might be part of larger complexes (Extended Data Fig. 5a).

Moreover, we observed a hierarchy of structural evolution. Enzymes with more conserved surfaces tended to be less costly. This effect, although less strong, was also present for core residues, but was markedly diminished for binding sites, suggesting that surface residues are the primary sites for cost optimization (Fig. 4b). Moreover, the more expensive aromatic amino acids, such as phenylalanine and tyrosine, were more frequent in the variable and low-abundance enzymes. By contrast, the conserved, high-abundant enzymes contained higher amounts of the least expensive amino acids, glycine, glutamate and alanine (Fig. 4c,d and Extended Data Fig. 5b,c).

Addressing the molecular level, we detected a significant association between CR and amino acid content in at least one structural feature, for 11 of the 20 canonical amino acids ($P_{adj.} < 1 \times 10^{-4}$; Fig. 4e and Extended Data Fig. 5d). For instance, surfaces and coil regions of highly conserved enzymes had higher glycine content (Fig. 4f), whereas the core and helical regions had higher alanine content. Presumably, these small and inexpensive amino acids can replace many other amino acids, and their

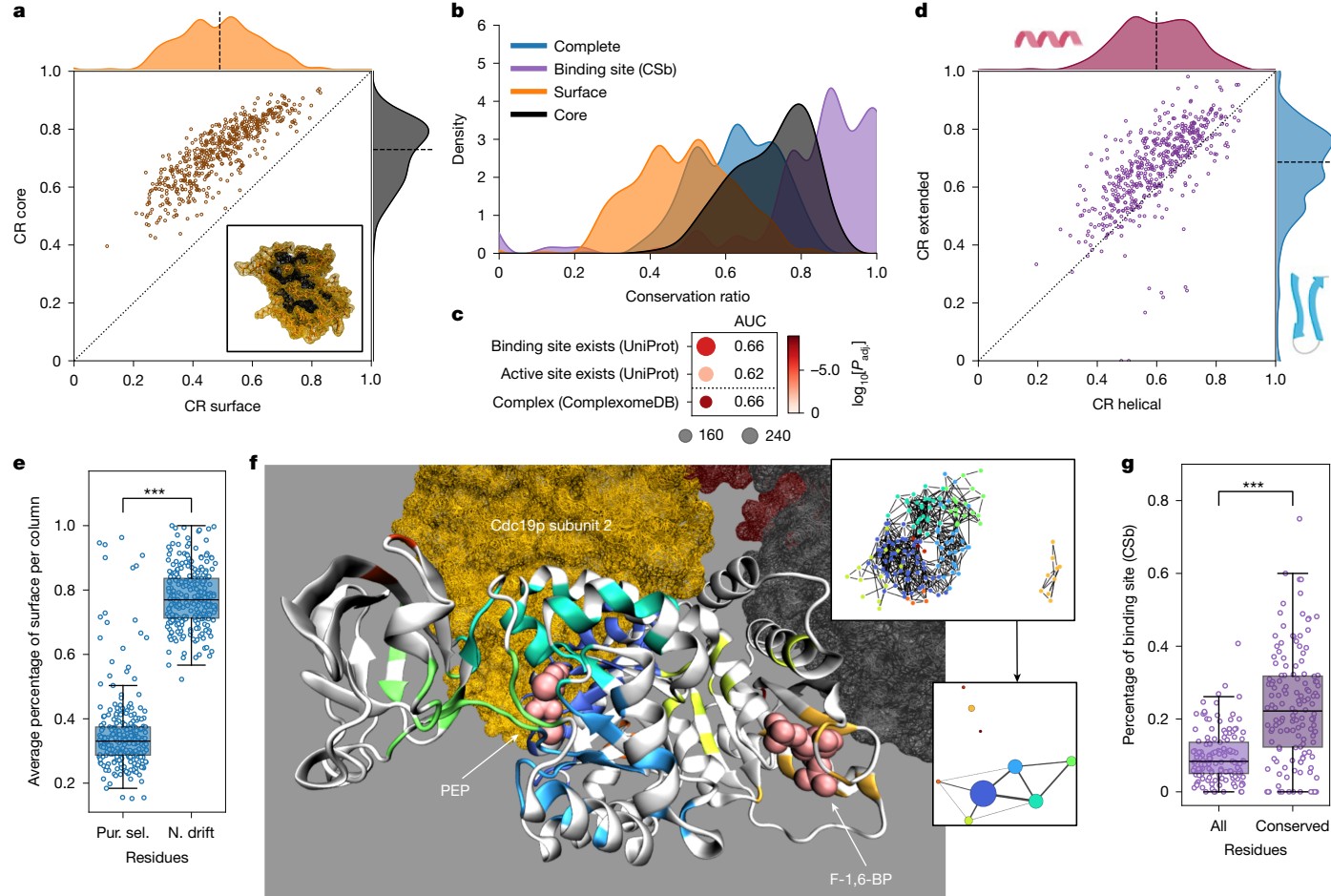

**Fig. 5 | Structural features and enzyme function are linked to conservation.** **a**, Mean conservation ratios of the core and the surface. Dotted line denotes the identity line and the dashed line denotes the axis median. Bottom right, depiction of the core (black) and surface (orange) of the protein based on relative solvent-accessible surface area for Fau1p. **b**, Distribution of the mean conservation ratio for all mapped residues (blue), the core (black), the surface (orange) and the binding-site residues (purple, CSb). **c**, Enrichment of orthogroups containing the 25% most conserved enzymes for structural features and catalytic properties. Dot size indicates the number of metabolic enzymes associated with the relevant list in our selection, colour indicates the adjusted $P$ value. **d**, Mean conservation ratios of the extended and the helical parts. Lines as in **a**. **e**, Percentage of residues in a column on the surface of the related structure for residues under purifying selection (Pur. sel.) or neutral drift (N. drift) (site model). ***$P < 1 \times 10^{-4}$, two-sided

Wilcoxon signed-rank test. **f**, Depiction of the conserved regions in Cdc19p. Each non-white region denotes a fully conserved cluster of amino acids. The spheres indicate the ligand (phosphoenolpyruvate (PEP), associated clusters: dark blue, light blue, green and cyan) and the activator (fructose-1,6-bisphosphate (F-1,6-BP), associated cluster: orange) (crystal structure: PDB 1A3W). Homotetrameric subunits are highlighted (associated cluster: dark blue). Top right, clustered network (nodes, residues; edges, Cα atoms within 10 Å; colours, cluster labels). Bottom right, summary of the clustered network (node sizes, adjusted cluster sizes; edges, connectivity; edge widths, edge number). **g**, Percentage of either all or only fully conserved residues identified as binding sites (CSb, $n = 140$). The line denotes the median and the boxes denote the first and third quartile, the whiskers extend up to 1.5 times the IQR. ***$P < 1 \times 10^{-4}$, two-sided Wilcoxon signed-rank test.

physicochemical properties are more suited to either the core or the surface. Furthermore, the surfaces of the variable enzyme structures had a higher cysteine content. This is presumably because cysteine often performs dedicated functions related to its high reactivity, such as catalytic activity or stabilization through disulfide bridges[38]. Notably, there were no significant associations between the presence of any amino acid in binding sites and the CR of the enzymes.

## Conservation of structural features
### Hierarchy of structural evolution

At the substructure level, the core of the enzyme was more conserved than the surface (Fig. 5a). Furthermore, the binding sites had the highest CR (Fig. 5b and Extended Data Fig. 6a), and orthogroups containing fewer variable enzymes were significantly enriched for possessing a known binding site (AUC = 0.62 and 0.66; Fig. 5c). An exception was the orthogroup of the fatty acid synthetase subunit Fas2p, in which

the binding site ($n = 21$ (UPb), 95 (CSb) amino acids) was more variable than the overall protein (Extended Data Fig. 6b,c).

At the level of the secondary structure, most helical portions (for example, α-helices) varied more than did extended portions (for example, β-sheets) (Fig. 5d). This result is robust to the tendency of AlphaFold2 to wrongly predict random coil regions (2.28%) as helical more than as extended[39] (Supplementary Note 2). To our surprise, helical portions were also more variable than mapped random coils, which are assumed to have a higher dynamic flexibility (Extended Data Fig. 6d and Supplementary Note 5). One explanation for this might be the presence of turns in the random coil regions, which show high conservation (Extended Data Fig. 7a,b).

Another factor that can influence structural evolution is larger architectures (protein folds). To assess this, we extracted fold information for those proteins in our dataset that were present in The Encyclopedia of Domains database[40]. Indeed, amino acids in these folds are slightly more conserved (two-sided Wilcoxon signed-rank test, $P < 1 \times 10^{-44}$,

adjusted Cliff's Δ = 0.64). Consistently, the two most dominant folds, the Rossmann fold (CATH fold 3.40.50) and TIM barrel (CATH fold 3.20.20), also exhibit higher conservation (two-sided Wilcoxon signed-rank test, $P < 1 \times 10^{-4}$, adjusted Cliff's Δ = 0.36 (Rossmann fold) and $P < 1 \times 10^{-7}$, adjusted Cliff's Δ = 0.86 (TIM barrel); Extended Data Fig. 6e). The orthogonal bundle (CATH architecture 1.10) was enriched in the most conserved enzymes (Fisher's exact test, $P_{adj.} < 1 \times 10^{-3}$).

### Conserved clusters capture interactions

To identify important functional residues, we conducted evolutionary selection analysis by estimating non-synonymous (dN) and synonymous (dS) substitution rates at both the whole-protein and site levels[41] (Extended Data Fig. 8 and Supplementary Note 6.1). We observed a higher probability of neutral drift (dN/dS < 1) in surface residues (Fig. 5e). Moreover, in specific cases with (1) low conservation and high whole-protein dN/dS values (ETC proteins and Pox1p); or (2) the presence of sites under positive selection across all species, for example the GAPDH orthogroup containing Tdh1/2/3p, we observed residues with evidence of positive selection in specific branches using a branch-site model[42,43]. In Pox1p, positively selected residues are found in the homodimer interface (Extended Data Fig. 9a,b), whereas in GAPDH, positively selected residues were primarily on surface-exposed residues, and did not affect the multimeric protein–protein interaction (PPI) sites. Moreover, one positively selected residue in GAPDH is within distance of the substrate–cofactor binding interface (Extended Data Fig. 9c,d). Finally, in the ETC supercomplex, 96 out of 157 unique residues with signatures of positive selection were on internal PPI sites (Extended Data Fig. 9e–g). Thus, despite surfaces being most divergent, followed by core residues and low diversification in the substrate binding sites, positive selection can occur in all structural elements (Supplementary Note 6.2).

In parallel, we noticed that fully conserved structural regions are often organized in clusters. Generating a network of fully conserved residues, resulted in a median number of 4.00 (IQR = 2.00–6.00) clusters with a median size of 18.55 amino acids (IQR = 14.00–23.64) per cluster (Supplementary Table 5.1). For example, in the pyruvate kinase orthogroup (Cdc19p of *S. cerevisiae*), these clusters correspond to the allosteric activation site for fructose-1,6-bisphosphate, the substrate-binding site for phosphoenolpyruvate and to a symmetric PPI site (Fig. 5f). We thus speculated that these clusters could correspond to metabolite-binding or other interaction sites. Indeed, they contain twice as many substrate- and ligand-binding-site residues (Fig. 5g and Extended Data Fig. 6f). Moreover, 91% (CSb) and 97% (UPb) of the annotated binding sites overlapped with at least one cluster. On the other hand, 27% (CSb) or 50% (UPb) of these clusters did not overlap with annotated binding sites. To test whether clusters can be identified as containing known binding sites, we trained a histogram-based gradient boosting classification tree on physicochemical properties. Evaluating ten fivefold cross-validations, we obtained an average balanced accuracy of 0.63 and average AUC of 0.68 on test data removed before training (Extended Data Fig. 6g), outperforming two random models (balanced accuracy = 0.57, AUC = 0.60 (random binding site) and balanced accuracy = 0.53, AUC = 0.54 (randomized labels); Extended Data Fig. 6h). Projecting the prediction confidence onto individual amino acids confirms that residues in known binding sites are more likely to be predicted as part of a binding site ($P < 1 \times 10^{-38}$, two-sided Wilcoxon signed-rank test, adjusted Cliff's Δ = 0.78). Moreover, we tested the degree to which these clusters can correspond to other types of interaction site and observed a slight over-representation of known PPI sites as extracted from the RCSB PDB (Extended Data Fig. 6i; Cliff's Δ = 0.23 (compared with small-molecule-binding sites with 0.84 (CSb) and 0.92 (UPb)), which is consistent with a high conservation of PPI sites (Extended Data Fig. 6j). In agreement, enzymes with more physical PPIs (Extended Data Fig. 6k; Kendall $\tau = 0.32$; $P_{adj.} < 1 \times 10^{-24}$) and enzymes involved in protein complexes[44] (ComplexomeDB; Fig. 5c)

were more highly conserved. Thus, although highly conserved clusters are dominated by small-molecule-binding sites, they also reflect PPI sites. Notably, some known small-molecule-binding sites were not in highly conserved clusters. This situation might be explained by the small size of some small-molecule-binding sites (such as metal interaction sites), but also points to the need for better annotation of existing sites.

## Discussion

The metabolic network is evolutionarily ancient, and its origins are commonly explained by two prevailing hypotheses. One suggests that its topological organization emerged as a consequence of enzyme evolution, whereas the other proposes that the metabolic network structure originated from non-enzymatic reactions[45]. Although metabolic evolution has probably experienced elements of both, increasing amounts of experimental evidence favour the second scenario. For example, many enzyme-catalysed reactions resemble non-enzymatic reactions as promoted by metal ions found frequently in Archaean sediment[46,47]. Moreover, despite considerable divergence in the enzyme sequences, the basic structure of the metabolic network remains conserved[3,48]. At the same time, modern metabolic pathways are highly efficient and respond to the environment, which suggests that they are optimized during evolution[49].

Studies into enzyme evolution support the model in which metabolic pathways evolve alongside chemical topologies, followed by their evolutionary optimization. Comparative genomics and detailed structural and functional investigations have described enzyme evolution as a dynamic process shaped by genetic innovations, biological constraints, cost and ecological interactions[5,50–52]. Functional clustering of enzymes and their domains into families has provided insights into the extent of gene duplication and divergence across the tree of life[40,53–56]. Furthermore, studies on niche adaptation have demonstrated how metabolic capabilities are lost and gained through both the expansion of gene families and the functional diversification of promiscuous enzymes[2,18,19]. These findings suggest that enzyme evolution follows selective pressures that balance catalytic efficiency with cellular metabolic demands and resource allocation.

We speculated that the ability to generate protein structures systematically and across species barriers[1] enables the integration of high-resolution structural data into functional genomic approaches, aiding the understanding of evolutionary processes. Such an approach is certainly constrained by the accuracy of structural prediction[39,57]. We focused on 26 diverse species selected from the Saccharomycotina subphylum to build on well-annotated genome sequences, enzyme functional annotations, genome-scale metabolic network reconstructions and proteomic data. The 11,269 enzyme structures examined cover most metabolic pathways, GO terms and enzyme classes present in their metabolic networks. We asked whether metabolic properties that differ between species, pathways and enzyme classes can help to explain the range of sequence diversity in structurally conserved regions observed in the different orthogroups. In this way, we identified metabolic constraints that influence structural evolution at different biological scales, from the level of the organism to those of pathways, individual enzymes and enzyme substructures. We also identified pathways, enzyme classes and, in some instances, structural elements that contribute to these relationships.

At the scale of the organism, we found that structural evolution is influenced by niche specialization, such as nutritional preferences. This included changes from oxidative to fermentative metabolism, which are a dominant metabolic module in yeast, other microorganisms and higher organisms. Changes from fermentation to respiration represent a major metabolic shift as fermentation is faster and less costly in terms of resource allocation, but oxidative metabolism has a better stoichiometry for ATP production, and imposes constraints on antioxidant metabolism[13,58,59].

At the pathway scale, our data suggest that enzyme structural evolution depends on pathway membership, the type of reaction catalysed and interactions with other metabolites. Enzymes involved in central carbon metabolism, oxidoreductases and metal-binding enzymes were the most constrained, whereas hydrolases and enzymes of more peripheral metabolic pathways, such as those functioning in lipid metabolism or protein glycosylation, diversified. Interestingly, we report an interdependency between enzyme conservation, metabolic flux and processivity, but find that variability of flux and processivity was more important than the total amount, indicating for these properties that the dynamic nature of metabolism constrains enzyme evolution more than do static properties.

At the structural level, our study confirmed that high-abundance enzymes evolve towards a more cost-efficient amino acid composition[14–16] with cost optimization depending on enzyme class and structural element, and distinct amino acid substitutions prevailing in different contexts. Most optimization occurs on surface regions, with the least in binding sites. Furthermore, structural features exhibit specific trends in amino acid substitutions. For example, alanine is more prevalent in the core, whereas glycine residues are more common on the surface of highly conserved enzymes.

It has been proposed that the diversification of enzyme structures can lead to new metabolic capabilities[4,5,55]. Our dataset is consistent with this possibility but highlights that most of the structure-driven diversification results in shifts between chemically similar reactions. Our dataset identifies no example of a higher order change in enzyme function, and at the molecular level, binding sites are highly conserved and not optimized for costs. Therefore, we observe the formation of clusters of high structural conservation in small-molecule-binding sites. Our data suggest that these clusters can be used to annotate previously undescribed binding sites, whereas some serve other functions such as PPI.

Across all our analyses, we obtain a consistent picture that the relationship between structure and evolutionary constraints is dominated by catalytic function. Notably, hydrolases differ in several properties (Extended Data Fig. 10a–e) and escape most of the otherwise highly pronounced relationships. This is not due to overall structural differentiation, as the MR was not significantly different to other enzyme classes (Extended Data Fig. 10f), neither was it due to changes in binding site conservation (Extended Data Fig. 10g,h). We speculate that a contributing factor to this situation is that hydrolases do not require a cofactor[60], that their reaction mechanism is thus less constrained in evolution, and that they participate in a diverse spectrum of metabolic processes. By contrast, the high conservation of oxidoreductases is predominantly explained by their role in central metabolism.

Overall, this dominance of the catalytic constraints across all layers investigated is consistent with a model in which metabolic enzymes evolve alongside the chemical topology of the metabolic network, with structural components involved in catalysis changing the least. An alternative hypothesis, that constraints on enzyme structure drive changes in metabolism, would result in more flexible binding sites in conserved structures. This notion does not rule out that structural changes in enzymes could cause changes in substrate specificity and amplification of promiscuous activities that could pave the way for the evolution of metabolic pathways, for instance, through the selection of a promiscuous reaction. These findings illuminate the relationship between enzyme function, metabolic environment and structural evolution, providing innovative strategies for enzyme annotation and metabolic network engineering.

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

## Reporting summary

Further information on research design is available in the Nature Portfolio Reporting Summary linked to this article.

## Data availability

Protein expression levels (raw data are available via ProteomeXchange with identifier PXD064343), predicted structures, alignments and other data are available at Figshare (https://doi.org/10.6084/m9.figshare.29117966)[61]. Supplementary tables are included in the Supplementary Information. Source data are provided with this paper.

## Code availability

Functions used to map the structures and visualize the data are available at GitHub (https://github.com/OliverLemke/structure_comparison (including an example of how to use the functions) and https://github.com/OliverLemke/basic_plots). Functions used to assemble MSAs, phylogenetic trees and to do evolutionary selection analysis are available at GitHub (https://github.com/heineike02/diverse_yeast). Additional code is available at Figshare (https://doi.org/10.6084/m9.figshare.29117966)[61].

61. Lemke, O. et al. The role of metabolism in shaping enzyme structures over 400 million years. *Figshare* https://doi.org/10.6084/m9.figshare.29117966 (2025).

**Acknowledgements** We thank J. Bahler, J. Yu, J. Muenzner, J. Hartl, C. Jakobsen, T. Alam, M. White, S. Alvarez-Carretero and Z. Yang for scientific advice; J. Tamanini for editing support and L. Szyrwiel, D. Ludwig and M. Muelleder for preparation and support for the proteomics measurements. O.L. and N.C. are supported by the European Research Council (ERC) under grant agreement ERC-SyG-2020 951475 awarded to J.B. and M.R. B.M.H. and S.K.A. are supported by Wellcome Trust grant (IA 200829/Z/16/Z). A.Z. is supported by Biotechnology and Biological Sciences Research Council (BBSRC) grant number BB/Y000730/1, Marius Jakulis Jason Foundation, Swedish Research council (Vetenskapsrådet) grant no. 2023-04254, 2019-05356, Formas grant 2019-01403. The data handling and computations were enabled by resources provided by the National Academic Infrastructure for Supercomputing in Sweden (NAISS), partially funded by the Swedish Research Council through grant agreement no. 2022-06725. The computations for structure predictions were enabled by the Berzelius resource provided by the Knut and Alice Wallenberg Foundation at the National Supercomputer Centre. J.L.S. is a Howard Hughes Medical Institute Awardee of the Life Sciences Research Foundation. Research in the laboratory of A.R. is supported by the National Science Foundation (DEB-2110404), the National Institutes of Health/National Institute of Allergy and Infectious Diseases (R01 AI153356) and the Burroughs Wellcome Fund.

**Author contributions** O.L., B.M.H., T.I.G. and M.R. conceptualized the work. O.L., B.M.H., S.V., N.C., J.L.S. and T.I.G. developed the methodology. O.L., B.M.H., S.V., N.C., J.L.S., L.S. and F.L. implemented the software for the analysis. O.L., B.M.H., S.V., F.L., F.A. and C.T.L. validated the data. O.L., B.M.H., N.C., J.L.S. and L.S. performed the formal analysis and investigation of the data. S.V. generated the structures. A.Z. and M.R. provided resources. O.L., B.M.H., S.V., N.C., J.L.S., L.S., F.L., F.A. and S.K.A. were involved in data curation. O.L., B.M.H. and M.R. wrote the original draft. O.L., B.M.H. and M.R. revised the manuscript. O.L. and B.M.H. prepared the figures. T.I.G. and M.R. supervised and O.L., B.M.H. and M.R. administered the project. J.N., A.R., A.Z., J.B. and M.R. provided funding or resources to at least one co-author.

**Funding** Open access funding provided by Max Planck Society.

**Competing interests** M.R. and A.Z. are founders and shareholders of Eliptica. J.L.S. is an adviser for ForensisGroup. J.L.S. is a scientific consultant for FutureHouse. Over the course of this project, J.L.S. was a Bioinformatics Visiting Scholar at MantleBio. A.R. is a scientific consultant for LifeMine Therapeutics. The other authors declare no competing interests.

**Additional information**
**Correspondence and requests for materials** should be addressed to Markus Ralser.

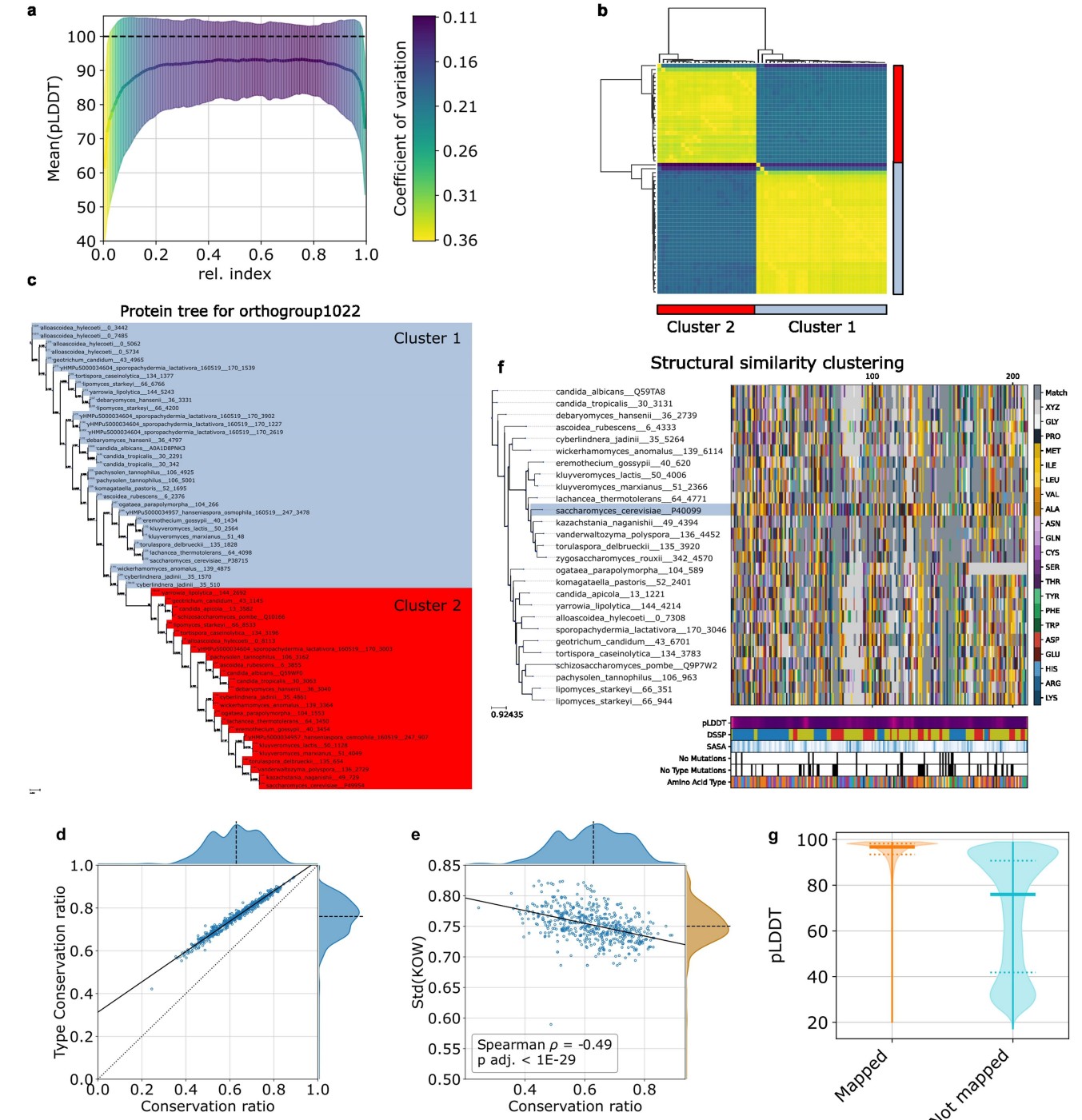

**Extended Data Fig. 1 | The predicted protein structures are well-structured, but small orthogroup refinements are required.** a) Mean pLDDT score as calculated by Alphafold2 versus the relative, length-normalised amino acid index for all our simulated structures, divided into 100 bins, due to varying protein lengths. Error bars denote the standard deviation coloured by the coefficient of variation. b + c) Structural alignment-based orthogroup refinement illustrated on orthogroup 1022. b) Hierarchical clustering of structures using US-align scores. The two resulting clusters are highlighted. Cluster 1 (blue) contains the reference structure *S. cerevisiae* Gre3p (Uniprot: P38715) and is named OG1022_REF_Scer_AF-P38715-F1-model and cluster 2 (red) contains Nit3p (Uniprot: P49954) and is named OG1022_REF_Scer_AF-P49954-F1-model. c) Phylogenetic tree based on US-align structural alignment of the entire original orthogroup 1022. The clusters based on hierarchical clustering correspond to structures that cluster together in the

protein tree. d) Mean CR versus mean amino acid type CR. The dotted line denotes the identity line, other lines as in e). e) Mean CR versus variability of the octanol-water partition coefficient (KOW). Solid line is the best linear fit and the dashed line denotes the axis median. f) Example of the structural alignment with respect to the reference structure. In addition, in the bottom panel other features, like pLDDT, solvent-accessible surface-area (SASA), secondary structure prediction (DSSP), fully conserved residues and fully conserved amino acid types are depicted. Colour according to amino acids. In case of light grey, no residue could be aligned to the reference. A dark grey colour indicates an agreement with respect to the reference structure for the respective amino acid. The more dark grey, the higher the conservation ratio. g) Maximum-normalised violin plot of the pLDDT for the density distribution containing every single residue grouped by residues that could be mapped (*n* = 5,527,262) and that could not be mapped (*n* = 1,454,754) respectively.

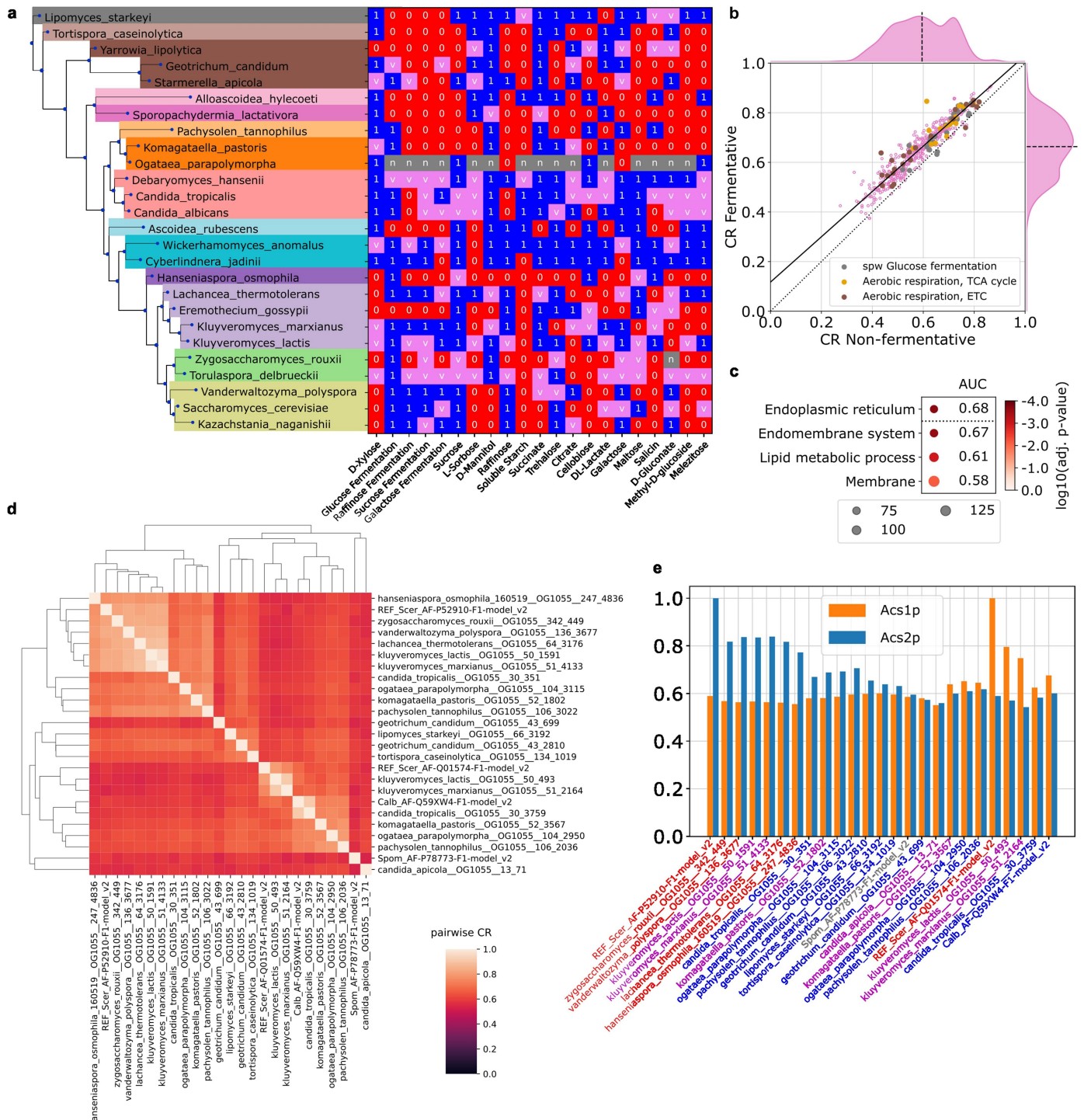

**Extended Data Fig. 2 | Nutrient utilization constraining structural evolution in *Saccharomycotina*.** a) Nutrient utilization for different carbon sources according to Kurtzman et al., 2011[17,18]. 1 denotes growth, 0 denotes lack of growth, v denotes variable growth, and n denotes missing data. b) Mean conservation ratio per protein calculated for species that ferment glucose versus species that do not ferment glucose in Kurtzman et al.[17]. Pathways related to aerobic respiration or anaerobic fermentation are highlighted. The dotted line denotes the identity line, the solid line denotes the linear fit and the dashed line denotes the axis median. c) Significant GO slim enrichments of

the first quartile of the orthogroups containing proteins that show the largest differences in mean conservation ratio between species that ferment and species that do not ferment glucose. Size indicates the number of genes associated with the GO slim term in our selection, and colour indicates adjusted p-value. In addition the area under the receiver operating characteristic curve (AUC) is shown. d) Average-linkage hierarchical clustering of the pairwise CRs for the orthogroup containing Acs1p and Acs2p in *S. cerevisiae*. e) CR of Acs1,2p orthologues with respect to Acs1p/Acs2p. The labels are colored according to the capability of D-Xylose utilization as shown in A.

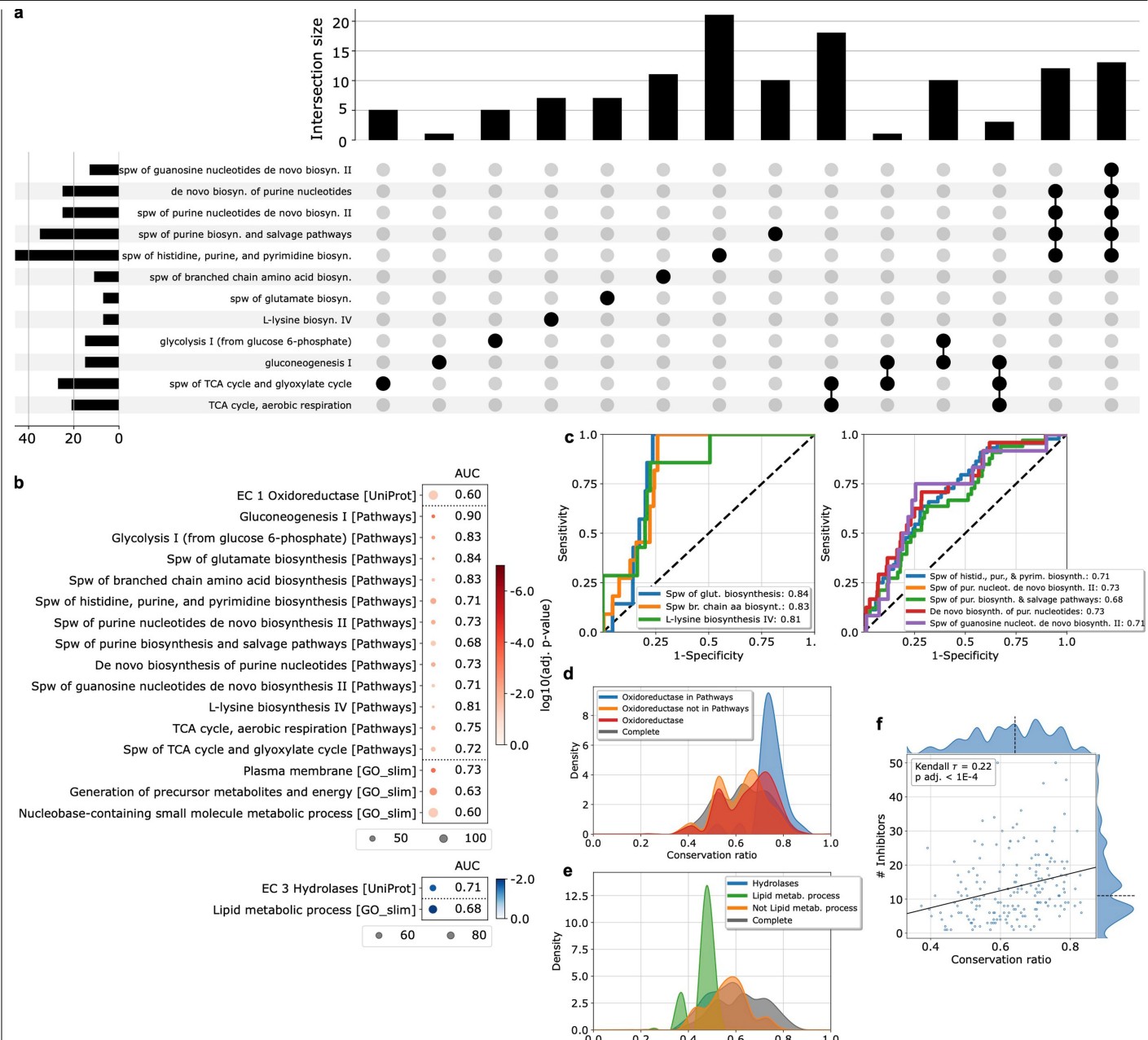

**Extended Data Fig. 3 | Metabolic factors constraining structural evolution in *Saccharomycotina*.** a) Upset-plot of the significantly enriched pathways, ordered by group dependency as shown in (b). b) Enrichment of the orthogroups containing the 25% most conserved (red, upper panel) and the 25% most diverse enzymes (blue, lower panel) for metabolic pathways from the yeast pathways database and GO slim terms. Size indicates the number of genes associated with the term or pathway in our selection, and colour indicates adjusted p-value. In addition, the AUC is shown. Super pathway is abbreviated as "Spw". c) ROC-curves for highly enriched pathways. The number in the legend indicates the AUC. The dashed line denotes the identity line representing random sampling.

d) Distribution of the mean conservation ratio of orthogroups assigned to oxidoreductases (red). In addition, the distributions for oxidoreductases that are (blue) or that are not (orange) members of the enriched pathways from (c) and Fig. 2d are depicted. The average distribution for all orthogroups is shown in black. e) Distribution of the mean conservation ratio of orthogroups assigned to hydrolases (blue). In addition, the distributions for hydrolases that are (green) or that are not (orange) members of the GO-term "lipid metabolic process" are depicted. The average distribution for all orthogroups is shown in black. f) Mean conservation ratio versus the number of inhibitors (*n* = 182). Solid line is the best linear fit and the dashed line denotes the axis median.

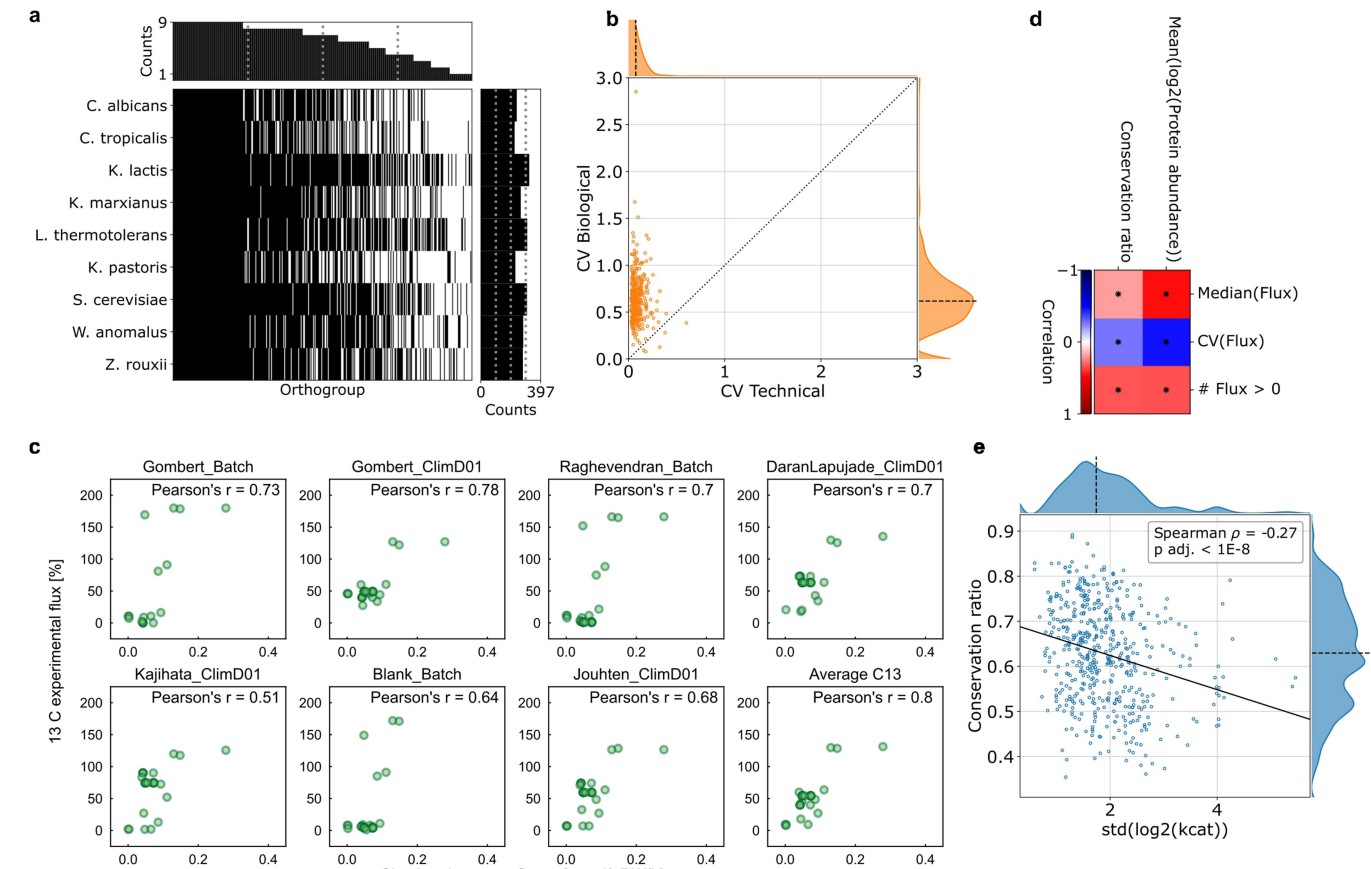

**Extended Data Fig. 4 | The proteomics data are of high-quality. Metabolic properties influence structural evolution.** a) Completeness of proteomic measurements for the selected orthogroups. For 397 of the original orthogroups protein abundance was measured in at least one species. A black line indicates a measured protein abundance of the orthogroup in the respective species. The bar plots account for the row and column sum. Grey dotted lines indicate the first, second and third quartiles. b) Scatter plot of the technical coefficient of variation (quality control samples) versus the biological coefficient of variation (measured species). The dotted line denotes the identity line and the dashed line denotes the axis median. c) Median simulated flux compared to [13]C experimental flux in 7 different datasets. The final plot compares median simulated flux to the median [13]C experimental flux. Only reactions that had data from at least 5 studies were used. d) Spearman's correlation of median predicted flux, coefficient of variations and Kendall's τ correlation of the number of budding yeast species (*n* = 329) in which a flux>0 was predicted[18] versus mean protein abundance and mean orthogroup conservation ratio. A * indicates an adj. p-value < 0.05. e) Mean conservation ratio versus standard deviation of the log2-transformed averaged *kcat*. Solid line is the best linear fit and the dashed line denotes the axis median.

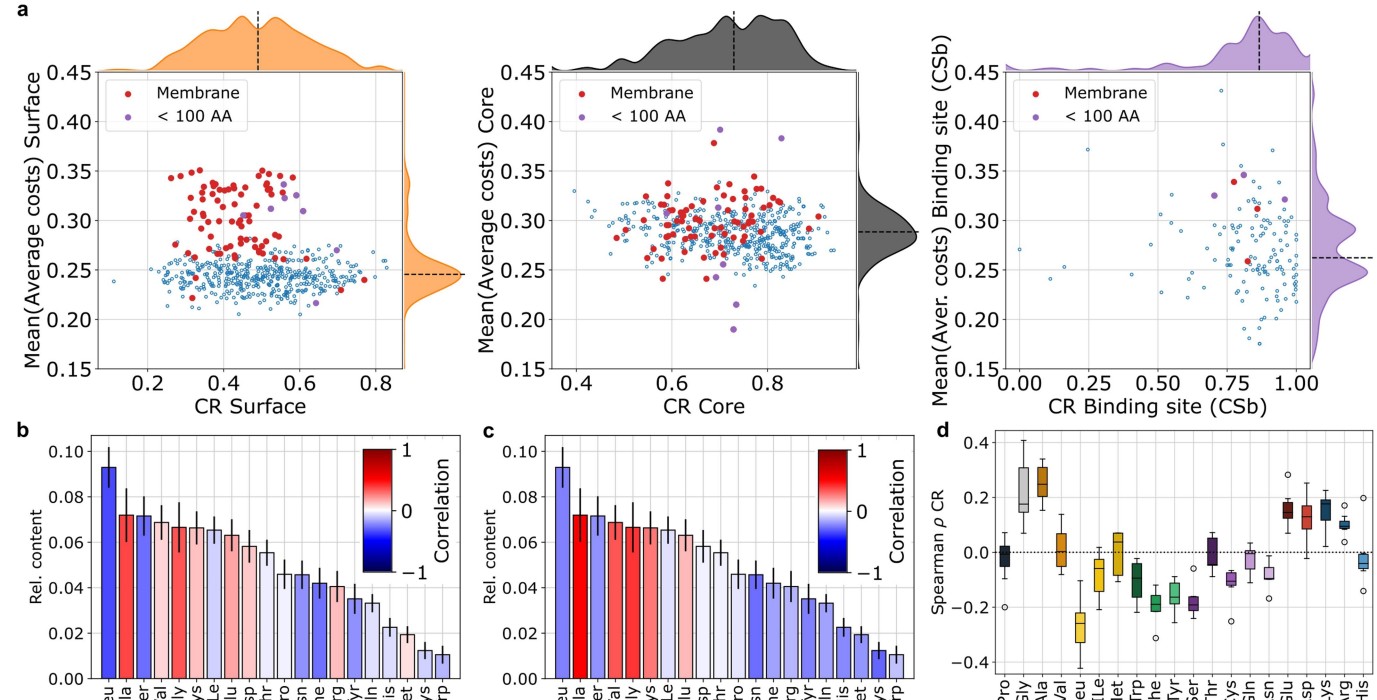

**Extended Data Fig. 5 | Amino acid costs are optimised depending on the structural feature and the physicochemical environment.** a) Conservation ratio of the surface (left), core (centre) and binding site (CSb) (right) versus their respective mean normalised costs. Membrane bound (red, $n$ = 84) and short protein chains (purple, $n$ = 8) are highlighted respectively. For the surface a bimodal cost distribution is observed. The dashed line denotes the axis median. b + c) Bar plot of the median relative amino acid content sorted in descending order. The error bars denote the MAD. The colouring of the bars indicates Spearman's correlation of the relative amino acid content of the entire protein versus b) the mean overall conservation rate and c) the log-transformed protein abundance. d) Boxplot of Spearman's correlation coefficients as shown in Fig. 4e (conservation ratio vs. relative amino acid content of different structural features). The line denotes the median and the boxes denote the first and third quartile, the whiskers in maximum 1.5-times the interquartile range. Outliers are shown as single dots.

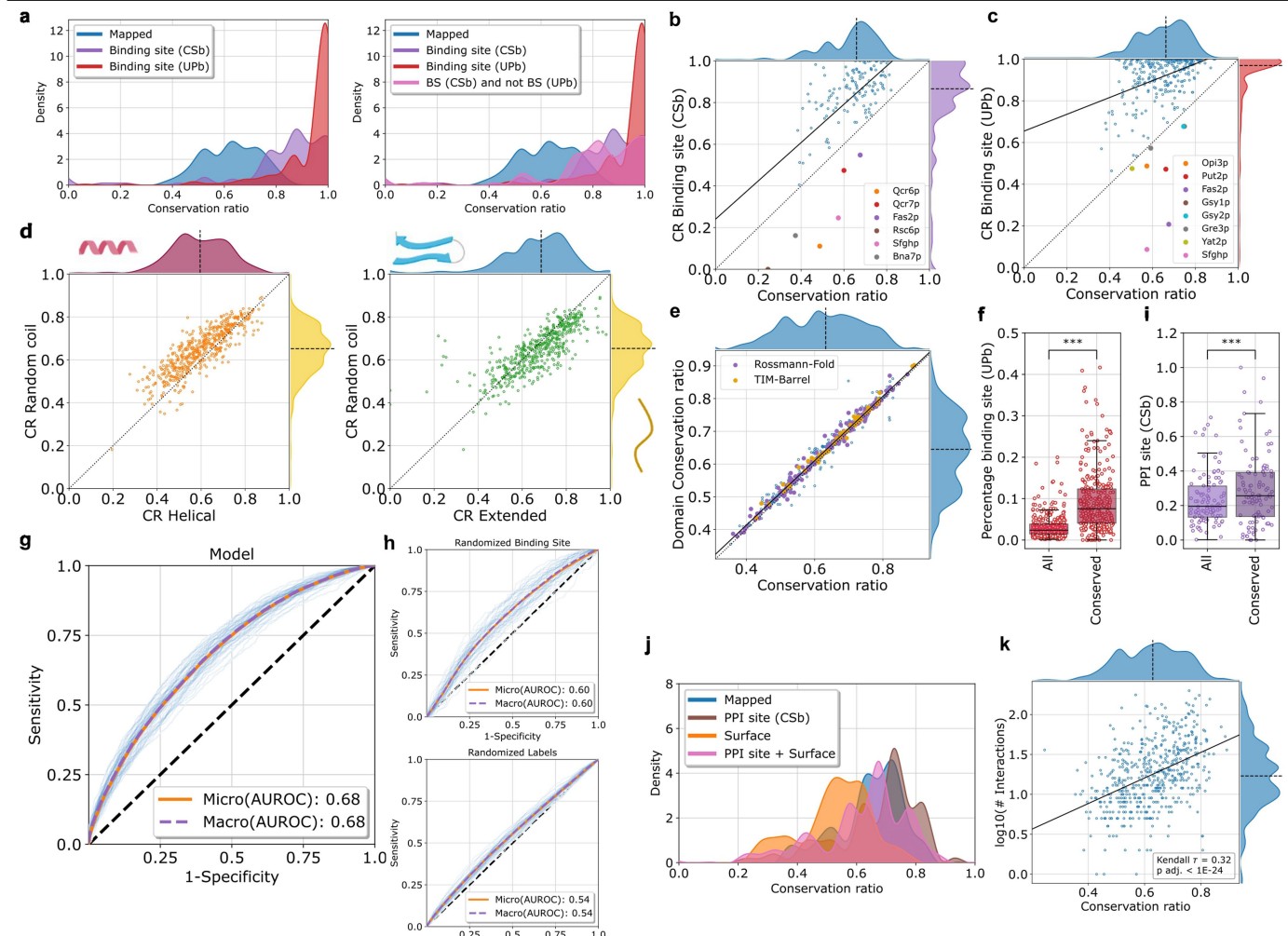

**Extended Data Fig. 6 | There is a hierarchy of evolution of structural features with surface residues being least constrained and binding sites the most.** a) Distribution of the mean conservation ratio for all mapped parts (blue) and only for the binding site (purple and red). The purple distribution depicts the CSb definition, the red distribution the UPb definition (left). In the right panel the pink distribution accounts for residues present in the purple distribution (CSb) and not present in the red one (UPb). b + c) Mean conservation ratio of the overall protein versus the conservation of the respective binding sites for b) the CSb definition and c) the UPb definition. Orthogroups below the dashed identity line are highlighted. Solid line is the best linear fit. Dashed line denotes the axis median. d) Mean conservation ratio of the random coiled versus mean conservation ratio of the helical parts (left) and the extended parts (right) respectively. Lines as in b). e) CR versus CR of known domains. Enzymes with the two most dominant folds are highlighted. Lines as in b). f + i) Percentage of either all or only fully conserved residues that are identified f)

as binding sites for the UPb definition (*n* = 317) of binding sites or i) as protein-protein-interaction (PPI) sites (*n* = 106). The line denotes the median and the boxes denote the first and third quartile, the whiskers extend up to 1.5-times the interquartile range. *** indicates a p-value < 1E-4, two-sided Wilcoxon signed-rank test. g + h) ROC-curves for a machine-learning model predicting known binding-sites. The performance of the test set, that was removed before training the model, for 50 runs as well as the micro and macro averages are shown. For comparison in g) the original labels and in h) randomized labels were used for training and testing. j) Distribution of the mean CR for all mapped parts (blue) and only for the PPI sites (brown) and the surface (orange) respectively. The pink distribution depicts surface residues that are part of a coordination site only. k) Mean CR versus the log10-transformed number of unique physical interactions taken from BioGRID via the Yeastmine interface (*n* = 525). Lines as in b).

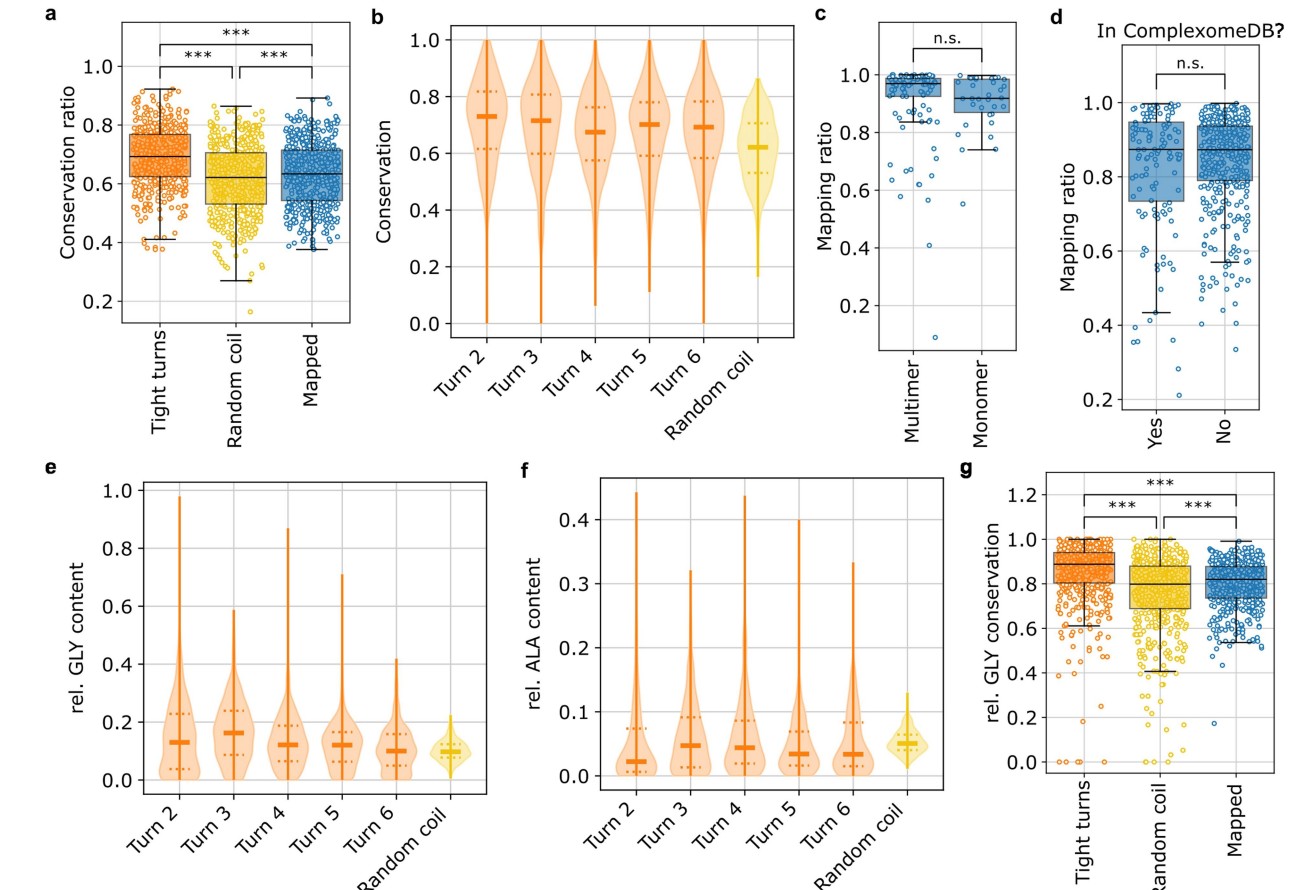

**Extended Data Fig. 7 | Our predicted structures capture multimeric structures well. Glycine and tight turns are highly conserved.** a) Boxplot for CR in tight turns, random coils or the whole protein. b) Distribution of CR for different turn lengths compared to random coils. c + d) Boxplot for c) MR of multimeric ($n$ = 106) and monomeric structures ($n$ = 34) and g) proteins in a complex as defined in ComplexomeDB ($n$ = 128) or not ($n$ = 401) performing two-sided Wilcoxon-Mann-Whitney test. *** indicates a p-value < 1E-4.

e + f) Distribution of relative e) glycine and f) alanine content for different turn lengths compared to all random coil residues. g) Boxplot for glycine conservation in tight turns, random coils or the whole protein. The line denotes the median and the boxes denote the first and third quartile, the whiskers extend up to 1.5-times the interquartile range. A *** indicates a p-value < 1E-4, two-sided Wilcoxon signed-rank test.

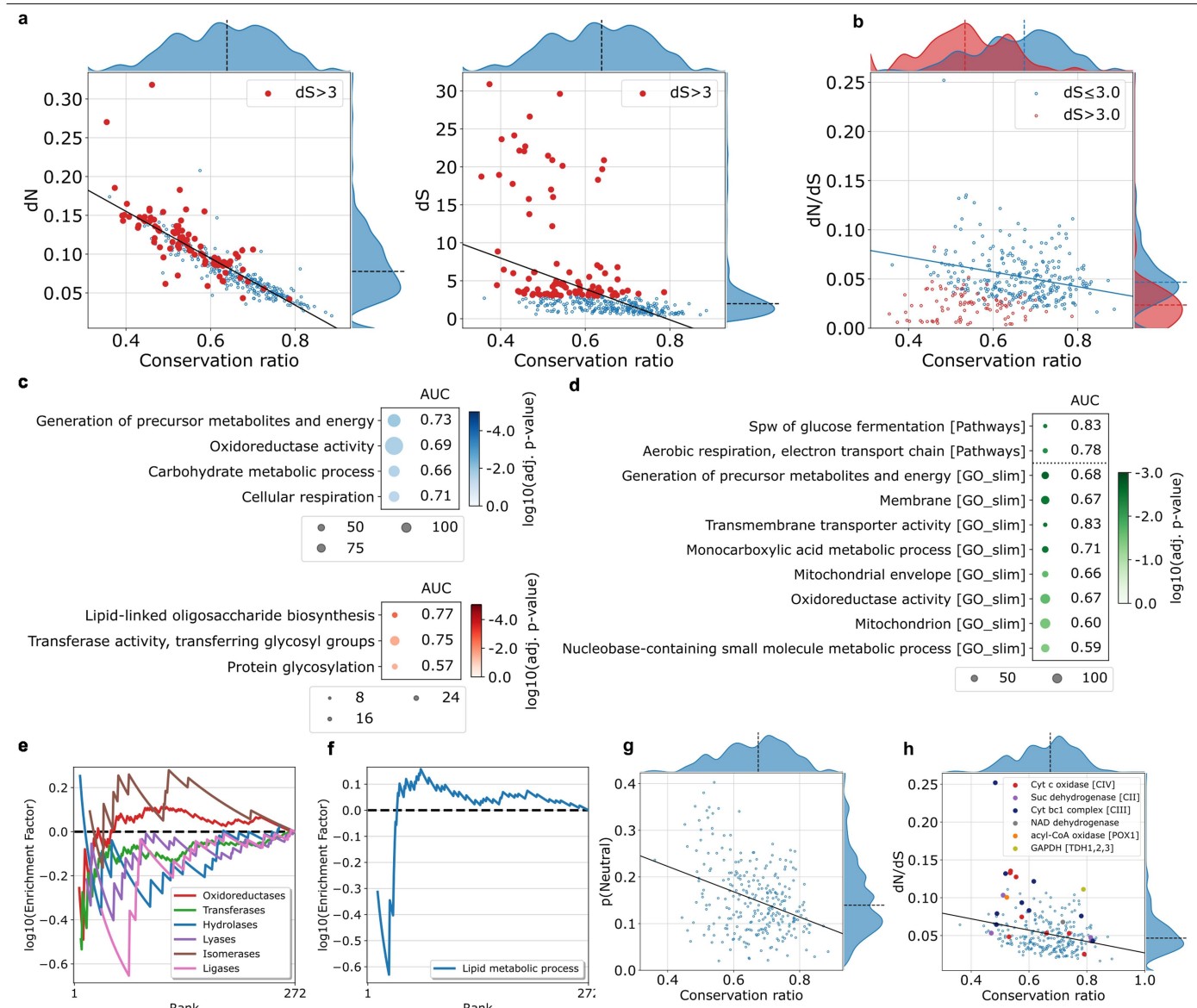

**Extended Data Fig. 8 | Refinement of the dN/dS calculation to remove skewed values.** a) dN (left) and dS (right) versus mean CR. Red dots denote a dS > 3, the solid line indicates the best linear fit and the dashed line denotes the axis median. b) dN/dS versus mean CR. Colouring according to a). The solid line indicates the best linear fit for the blue highlighted orthogroups and the dashed lines denote the axis median for the two groups. c + d) Significantly enriched pathways and GO slim terms of c) the blue (dS ≤ 3, *n* = 272) and red (dS > 3, *n* = 94) highlighted orthogroups with respect to the full set and d) orthogroups containing enzymes with the 25% highest dN/dS values. Dot size indicates the

number of genes associated with the term or pathway in our selection, and colour indicates adjusted p-value. In addition the AUC is shown. e + f) Enrichment factor for e) enzyme classes and f) the GO term "lipid-metabolic process" according to dN/dS. The dashed line indicates random sampling. g) Probability of a residue being under neutral drift is inversely correlated with CR. h) dN/dS versus the mean CR for orthogroups with dS < 3. Orthogroups that contain proteins of the electron transport chain as well as Pox1p and Tdh1/2/3p are highlighted. The solid line indicates the best linear fit and the dashed line denotes the axis median.

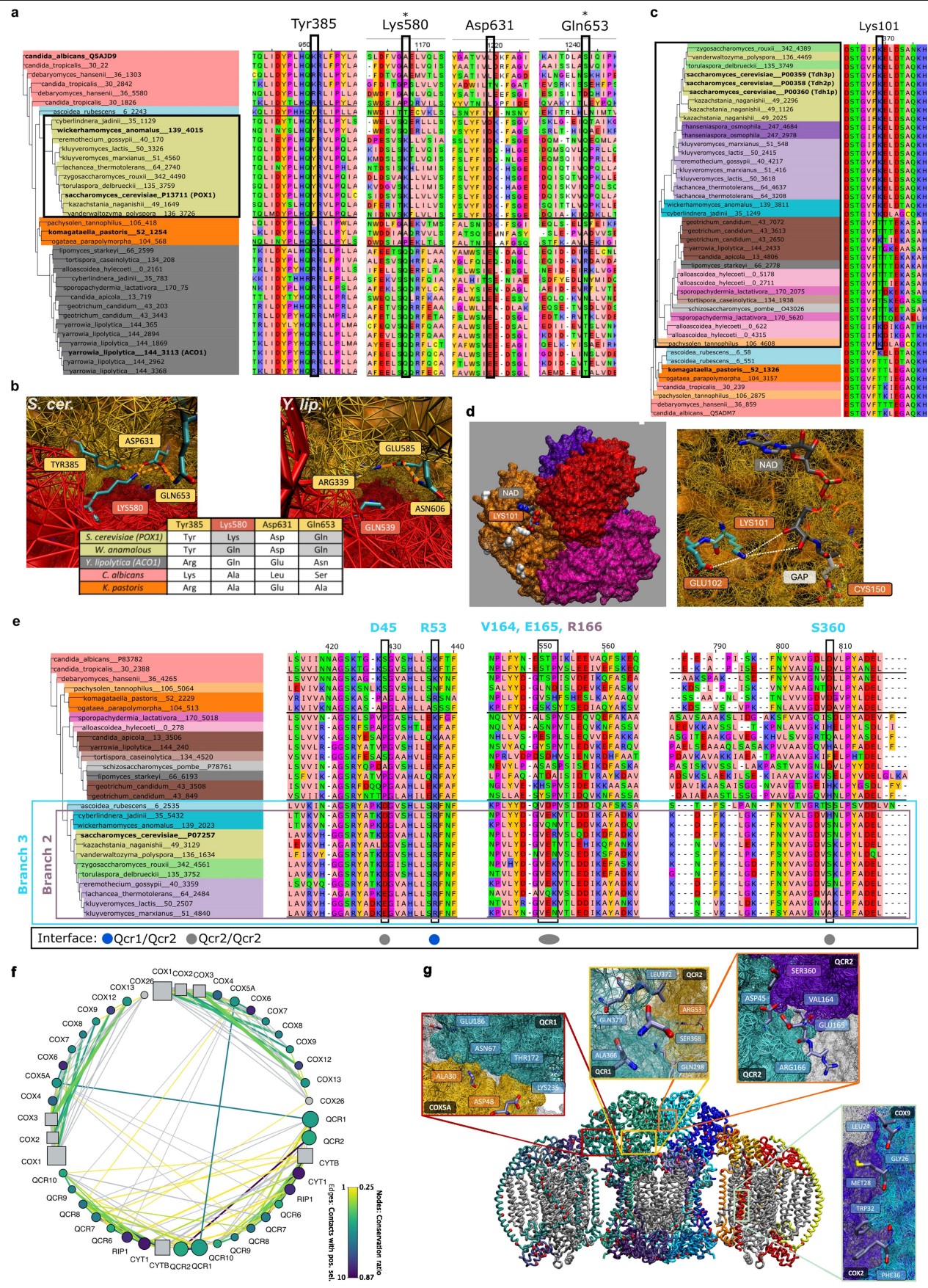

**Extended Data Fig. 9** | See next page for caption.

**Extended Data Fig. 9 | Positive selection in Pox1p, Tdh1/2/3p and the ETC.**
a) Multiple sequence alignment of selected regions of Pox1p orthologs. Asterix indicates residues under positive selection in the branch highlighted with the black box on the left. Other highlighted residues interact with positively selected residues in the structure. The dark square indicates the branch containing *S. cerevisiae* and *W. anomalous* orthologs for which residues under positive selection are highlighted in (b). b) Structure of the fatty-acyl coA oxidase homodimer interface obtained by using the crystal structure of the *Y. lipolytica* AOx protein, ylACO1 (right), (PDB:5Y9D). This crystal structure was also used as a reference to superimpose *S. cerevisiae* Pox1p structure from Alphafold2 (left). The positively selected amino acids Lys580 and Gln653 as well as the coordinating amino acids Tyr385 and Asn631 are highlighted. The red and the orange surfaces denote different protein chains of the homodimeric complex. The inset table denotes the residues present for various orthologs. Grey shading in the table denotes residues and branches that were under positive selection. c) Multiple sequence alignment of GAPDH enzymes in the region surrounding the K101 residue positively selected in the branch indicated by the dark square. d) Left: Tetrameric structure of GAPDH 3 from *S. cerevisiae* (PDB:3PYM) with positively selected residues highlighted in white. Right: Zoom into the binding site (predicted structure) with cofactor NAD and ligand GAP extracted from PDB:5JYA. The positively selected Lys101 as well as the coordinating Glu102 and the catalytically important Cys150 are highlighted. e) Multiple sequence alignment of Qcr2 highlighting residues detected as under positive selection in designated branches. f) Connectivity network of the cytochrome-c-oxidase/cytochrome bc1 complex (PDB: 6HU9). The node size indicates the length of the protein chain. Square nodes denote mitochondrially-encoded genes. The nodes are coloured with respect to the conservation ratio, where grey denotes missing orthogroups in our data selection. The edges indicate physical interactions between the proteins and edge width indicates the number of contacts between the surfaces of the two protein chains. The edges are coloured according to the number of positively selected amino acids from either protein chain on the surface between the two protein chains. g) Crystal structure of the cytochrome-c-oxidase/cytochrome bc1 complex. The structure consists of a homodimer made up of two ubiquinol cytochrome c reductase (complex III) complexes surrounded by two cytochrome c oxidase (Complex IV) complexes. Protein chains are coloured with respect to their conservation rate (left part) or chain ID (right part). Red dots denote positions where positive selection was detected in at least one branch. In addition, four interfaces are shown in detail, highlighting positively selected residues between the two surfaces. The respective positions in the complex are denoted in coloured boxes. Multiple sequence alignment illustrations created with Jalview.

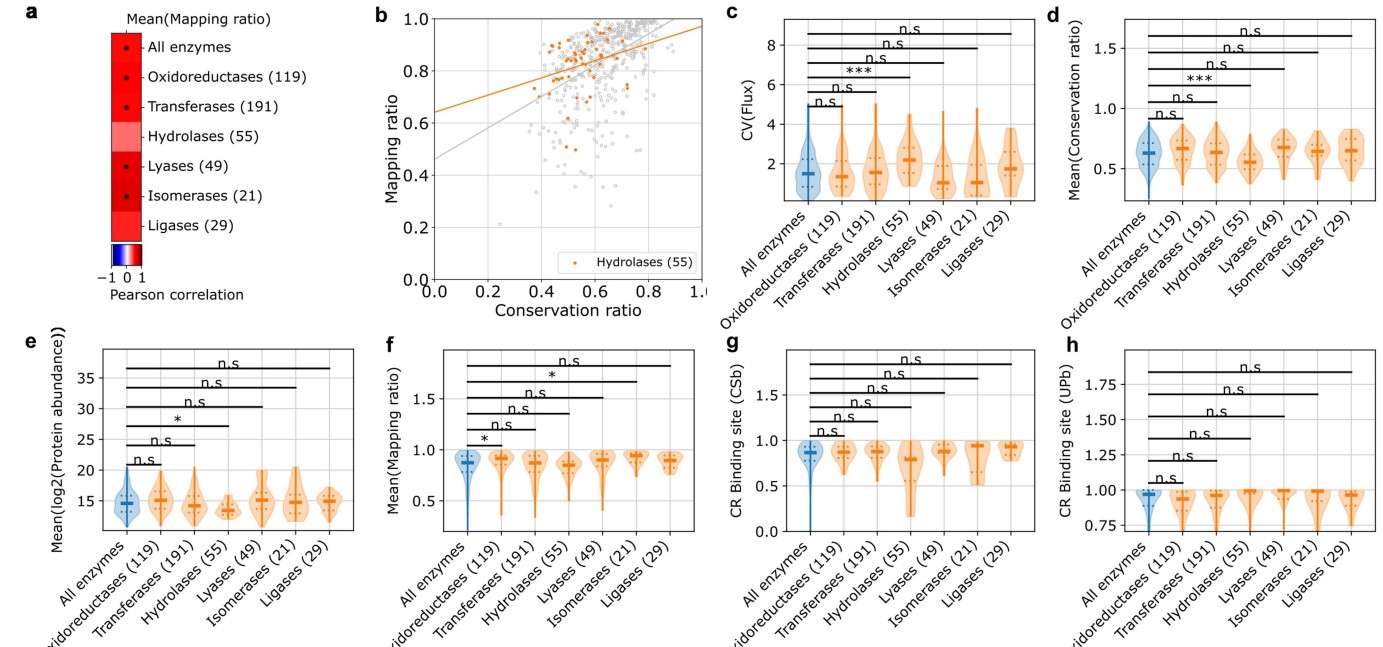

**Extended Data Fig. 10 | Hydrolases escape most relationships and differ significantly in several investigated properties.** a) Pearson correlation of the mean MR and the mean CR for different enzyme classes. A * indicates an adj. p-value < 1e-2. b) Mean CR versus mean MR with orthogroups containing hydrolases highlighted. The solid line denotes the best linear fit.

c-f) Distribution of c) flux coefficient of variation, d) CR, e) log2-transformed protein abundance and f) MR for different enzyme classes. A * indicates an adj. p-value < 1E-2, a *** an adj. p-value < 1E-4. g + h) CR of the binding site using g) the CSb and h) the UPb definition for different enzyme classes.

# Reporting Summary

## Statistics

For all statistical analyses, confirm that the following items are present in the figure legend, table legend, main text, or Methods section.

| n/a | Confirmed | |
|---|---|---|
| ☐ | ☒ | The exact sample size (*n*) for each experimental group/condition, given as a discrete number and unit of measurement |
| ☒ | ☐ | A statement on whether measurements were taken from distinct samples or whether the same sample was measured repeatedly |
| ☐ | ☒ | The statistical test(s) used AND whether they are one- or two-sided<br>*Only common tests should be described solely by name; describe more complex techniques in the Methods section.* |
| ☒ | ☐ | A description of all covariates tested |
| ☐ | ☒ | A description of any assumptions or corrections, such as tests of normality and adjustment for multiple comparisons |
| ☐ | ☒ | A full description of the statistical parameters including central tendency (e.g. means) or other basic estimates (e.g. regression coefficient) AND variation (e.g. standard deviation) or associated estimates of uncertainty (e.g. confidence intervals) |
| ☐ | ☒ | For null hypothesis testing, the test statistic (e.g. *F*, *t*, *r*) with confidence intervals, effect sizes, degrees of freedom and *P* value noted<br>*Give P values as exact values whenever suitable.* |
| ☒ | ☐ | For Bayesian analysis, information on the choice of priors and Markov chain Monte Carlo settings |
| ☒ | ☐ | For hierarchical and complex designs, identification of the appropriate level for tests and full reporting of outcomes |
| ☐ | ☒ | Estimates of effect sizes (e.g. Cohen's *d*, Pearson's *r*), indicating how they were calculated |

*Our web collection on statistics for biologists contains articles on many of the points above.*

## Software and code

Policy information about availability of computer code

| | |
|---|---|
| Data collection | Spectra deconvolution, protein identification and relative quantification for mass spectrometry was performed using DIA-NN 1.8 (Data-Independent Acquisition by Neural Networks) [Demichev V. et al., Nat Methods. 2020; 17(1): 41–44, doi: 10.1038/s41592-019-0638-x]. Usage details are provided in the methods section. |
| Data analysis | Functions used to map the structures and visualize the data are uploaded to https://github.com/OliverLemke/structure_comparison and https://github.com/OliverLemke/basic_plots. Functions used to assemble MSAs, phylogenetic trees and do evolutionary selection analysis are uploaded to https://github.com/heineike02/diverse_yeast. Further code used for data analysis and plot generation is available as a supplement. |

For manuscripts utilizing custom algorithms or software that are central to the research but not yet described in published literature, software must be made available to editors and reviewers. We strongly encourage code deposition in a community repository (e.g. GitHub). See the Nature Portfolio guidelines for submitting code & software for further information.

## Data

Policy information about availability of data

All manuscripts must include a data availability statement. This statement should provide the following information, where applicable:

- Accession codes, unique identifiers, or web links for publicly available datasets
- A description of any restrictions on data availability
- For clinical datasets or third party data, please ensure that the statement adheres to our policy

> PDB files for predicted structures and Proteomic data will be deposited on Figshare and access will be activated upon publication.

## Research involving human participants, their data, or biological material

Policy information about studies with human participants or human data. See also policy information about sex, gender (identity/presentation), and sexual orientation and race, ethnicity and racism.

| | |
|---|---|
| Reporting on sex and gender | NA |
| Reporting on race, ethnicity, or other socially relevant groupings | NA |
| Population characteristics | NA |
| Recruitment | NA |
| Ethics oversight | NA |

Note that full information on the approval of the study protocol must also be provided in the manuscript.

# Field-specific reporting

Please select the one below that is the best fit for your research. If you are not sure, read the appropriate sections before making your selection.

☒ Life sciences  ☐ Behavioural & social sciences  ☐ Ecological, evolutionary & environmental sciences

For a reference copy of the document with all sections, see nature.com/documents/nr-reporting-summary-flat.pdf

# Life sciences study design

All studies must disclose on these points even when the disclosure is negative.

| | |
|---|---|
| Sample size | NA |
| Data exclusions | We excluded orthogroups of predicted structures when they did not contain orthologs from a sufficient percentage of species. Parameters are defined in the Methods section. |
| Replication | NA |
| Randomization | NA |
| Blinding | NA |

# Reporting for specific materials, systems and methods

We require information from authors about some types of materials, experimental systems and methods used in many studies. Here, indicate whether each material, system or method listed is relevant to your study. If you are not sure if a list item applies to your research, read the appropriate section before selecting a response.

## Materials & experimental systems

| n/a | Involved in the study |
|-----|----------------------|
| ☒ | ☐ Antibodies |
| ☒ | ☐ Eukaryotic cell lines |
| ☒ | ☐ Palaeontology and archaeology |
| ☐ | ☒ Animals and other organisms |
| ☒ | ☐ Clinical data |
| ☒ | ☐ Dual use research of concern |
| ☒ | ☐ Plants |

## Methods

| n/a | Involved in the study |
|-----|----------------------|
| ☒ | ☐ ChIP-seq |
| ☒ | ☐ Flow cytometry |
| ☒ | ☐ MRI-based neuroimaging |

# Animals and other research organisms

Policy information about studies involving animals; ARRIVE guidelines recommended for reporting animal research, and Sex and Gender in Research

| | |
|---|---|
| Laboratory animals | N/A (ONLY YEASTS) |
| Wild animals | N/A |
| Reporting on sex | N/A |
| Field-collected samples | N/A |
| Ethics oversight | N/A |

Note that full information on the approval of the study protocol must also be provided in the manuscript.

# Plants

| | |
|---|---|
| Seed stocks | N/A |
| Novel plant genotypes | N/A |
| Authentication | N/A |

