## [Peer Review File · Nature]

Metabolism's Role in Shaping Enzyme Structures Over 400 Million Years

Corresponding Author: Professor Markus Ralser

Version 0:

Reviewer comments:

Referee #1

(Remarks to the Author)

Lemke and Heineike et al. leverage AlphaFold2 and a series of evolutionary and computational analyses to investigate the structural evolution of metabolic enzymes using 27 Saccharomycotina species. They propose that instead of structure dictating function, instead metabolism has shaped enzyme evolution via the topological organization of the metabolic network and the enzyme's molecular properties. While a compelling hypothesis is put forward, the authors never provide clear causal evidence that metabolism is driving the evolution of enzyme structure and not the inverse. While both scenarios are plausible, there are several alternate hypotheses to the ones the authors claim that are not sufficiently investigated. Additionally, the claims the authors make are not experimentally validated, or otherwise lack additional in silico experiments that are not already predicated on previous predictive models. We therefore suggest that this manuscript, while potentially impactful, is not acceptable for publication in its current form. While the study addresses an interesting topic, several significant issues must be addressed before publication. Moreover, there is some concern that, to rise to full impact, the manuscript would require experimental validation that is not feasible in a reasonable timeframe.

Fig. 1:

Many of the enzymes analyzed in this study operate as multimers. However, vanilla AlphaFold v2 (from the 2021 publication) is not tuned to predict multimeric structures. Did the authors investigate a combined approach using AlphaFold for monomers and AlphaFold-Multimer for multimeric enzymes? There should be a more thorough discussion of the biases between these methods, including those that arise in combining predicted structures from the two. Some of the variations discussed in enzyme classes could have resulted from poor predictions for enzymes that assemble as multimers, since base AlphaFold does a poor job of predicting a multimeric structure.

Fig. 2:

2a: The authors claim that enzymes from fermenting species showed higher structural conservation relative to *S. cerevisiae*. This claim is interesting only if it can be distinguished that this is not merely an effect of evolutionary proximity. For instance, the sampled species from the phylogenetic tree do not appear to be randomly distributed in evolutionary time. It is thus not clear if this apparent structural conservation is just a result of sampling bias. Are fermenting species closer in evolutionary time than they are to respiring species? The authors should also comment on the functional definition of fermentative vs. non-fermentative species as this makes a difference in the interpretation. Are the species labeled non-fermentative unable to ferment, or do they simply prefer not to? Many species can do both but have a strong preference for one. This is another potential bias that could change how these results are interpreted.

2c: One of the conclusions made from this figure is that hydrolases, which have more disordered regions, are less conserved in structure. However, there may be a trivial explanation here: hydrolases have a dramatically more straightforward and structurally localized mechanism, whereas oxidoreductases require binding of both a substrate and a large cofactor, necessitating a more complex structural solution, which is thus more likely to be evolutionarily constrained. If this is true, then there is seemingly little to be learned from these data. To be able to make the claim the authors wish to make, there are several alternative hypotheses that could be investigated.

2f: Given a large enough sample size, small differences in distributions can lead to a "significant" p-value. Although the

Wilcoxon-Mann-Whitney test indicates a significant difference, the data exhibit only a slight shift between groups, the quartile ranges are similar, and the median lines lie within the overlap of the quartile ranges, indicating no obvious difference in medians. The authors should include a calculation of effect size or other appropriate metric to determine if these differences are likely to be biologically meaningful.

Fig. 3:

3d: We are concerned with the authors' claims of predicting flux. Accurately estimating flux across species is likely to be highly inaccurate, biased by factors such as substrate availability, protein conformation, and more beyond the simple issues of protein steady-state abundance. The authors need to provide reasons to be confident that these models are accurately reflecting flux through these reactions. At a minimum, we might expect a simple glucose tracing experiment between strains measuring experimental flux through glycolysis or the TCA cycle compared to the flux predictions made using steady-state protein values from these pathways to determine the general accuracy of these flux predictions.

Fig. 4:

The concept of "cost" is quite important to the interpretations of the manuscript, but the concept seems quite opaque and somewhat arbitrary. More needs to be done to educate the reader on what is meant by "cost" and to convince them that it is being estimated appropriately. We find it hard to imagine that the cost of an amino acid is the same for all species, even within a defined subphylum, especially when they live very different lifestyles. If the cost for amino acids were different between species this would compromise the analysis.

Fig. 5:

5b: If a sub-domain could be either an alpha-helix or beta-sheet, it has been proposed that AlphaFold might have a bias to predict a helix preferentially over a sheet (<https://www.ncbi.nlm.nih.gov/pmc/articles/PMC9312937/>), so of course helices will end up being more variable. The authors should assess and address this potential concern with the AlphaFold model.

5e: In our opinion, the most impactful implication of this manuscript is the ability to predict a priori small molecule binding sites. While the authors show one interesting example here, this is not sufficient validation or meaningful follow-up of these claims, which seem to be a central claim of this work. More could be done to expand this meaningful outcome of the work.

Fig. 6:

It's not obvious from this figure or the text why there would be positive selection at protein interfaces. Would this be to increase or decrease affinity of the interaction? Perhaps, it could change the properties of the complex in some advantageous manner? The authors should help us understand the impact of this section of the manuscript. Again, this would require either experimental validation, or validation using additional data and not just predictions already predicated upon a prediction (e.g. the AlphaFold structure).

As an additional note, some of the explanations are at times overly complicated and difficult to follow. We acknowledge that this paper is computationally intensive and a sophisticated topic, but the authors could improve the accessibility. We appreciate that the analysis code and the code for the figures are shared. If the authors intend this work to be a reusable resource for bioengineers, as mentioned multiple times throughout the manuscript, we would recommend they consider publishing the code as interactive Jupyter notebooks with detailed documentation. This will enhance its usability and enable effective reuse.

Referee #2

(Remarks to the Author)

The article is very interesting with many novel observations. It describes a comprehensive and robust study of how metabolic processes have driven the evolution of enzyme structures over the last 400 million years. It is dense in interesting observations and yet overall mostly written in an accessible style with key findings summarized in the discussion. It is extremely timely benefitting from high quality AlphaFold structures that allow identification of structurally equivalent positions between relatives which are subsequently analysed for conservation of amino acid residue. This mapping enabled a large scale study correlating residue changes in enzymes with differences in various metabolic features. It also allowed the authors to analyse the evolutionary constraints acting on different regions in the enzyme structure. The study is impressive in its broad scope, analysing at different scales from pathways to enzymes and by exploiting the structural data, to enzyme substructures.

The authors selected a sound set of yeast fusion species from the Saccharomycotina subphylum for the study associated with a significant amount of experimental data. This allowed them to correlate what they describe as 'structural divergence' of enzymes within orthogroups with a wide range of interesting features including growth properties, protein abundance, variation in metabolic flux, sites under positive selection etc. The authors have used robust statistics throughout and highlight statistically significant observations.

They employ a careful selection of orthogroups ensuring these contain genes that cover a substantial proportion (40%) of the species analysed. They were able to generate a substantial number of high quality AlphaFold structures for the study, and it is particularly commendable that they exploited this structural data to refine the orthogroups.

Evolutionary variation is quantified by metrics that capture the mapping between enzymes and the proportion of mapped residues (superimposed on the reference structure within 2Å) that are identical. They use this conservation ratio as a proxy for structural divergence. It is measured against the reference *S. cerevisiae* structure in each orthologous group. I am not sure that 'structural divergence' is the most appropriate description of what they are analysing, but I discuss this more below.

Many of the trends the authors observe are not surprising and easy to grasp intuitively i.e. that enzyme sites implicated in function (e.g. interactions with substrates or proteins) are more highly conserved. However, there is a lot of fascinating detail throughout and a rather complex picture emerges involving a balance between multiple aspects (abundance, enzyme cost, functional constraints). I sometimes found it hard to keep track of the narrative as so many aspects are being considered. The figures are very good and helped a lot as did the discussion at the end, but I wondered if there were additional strategies that could help the reader keep track of the key findings as they emerge.

In summary, this is an extremely interesting and valuable study. A particularly useful aspect of the study is that it provides a comprehensive analytical framework that will be extremely valuable for other research groups studying or optimizing the properties of specific enzymes.

Minor Comments

1. The refinement of orthogroups using structure comparison and clustering was valuable given the correlation between structural similarity and functional similarity. It would be good to have a clearer understanding of the threshold used for this refined clustering, and how this had been decided. For example, do domains within the revised orthogroups cluster within 2 Ångströms? They mention the tool they used but it would be good if they cited the threshold in the main text and gave a widely used measure like RMSD, if possible.

Are all relatives in the final clusters likely to have the same enzyme chemistry and specificity or could there be divergence in these properties e.g. in specificity? It is known that more remotely related orthologs can diverge in function. Was there some evaluation of clustering threshold value based on structural similarity between enzymes with identical Enzyme Classifications at the 4th level (EC4), supporting identical chemistry and substrate specificity?

The clustering threshold is key to understanding whether the low CR of an orthogroup reflects a functionally identical set of relatives (same EC4) where residue changes are tuning functional or stability properties or a group of relatives in which there has also been some change in functional property e.g. change in the substrate acted upon.

2. Following on from that, to what extent are the relatives annotated? Since the authors examine whether specific types of enzymes (oxidoreductases, hydrolases) are less/more structurally divergent, I assume they have used experimental data from the Enzyme Classification or predicted annotations from UniProt to identify these types of enzymes. What proportion of the assignments are experimental? Do they see any variation in the functional annotations across orthogroups? Apologies if I missed a description of this in the supplementary. It would be helpful to have some mention in the main text, too.

3. I am not sure that structural divergence is a good term for what they are analysing. The metrics used are not based on the RMSD of superposed relatives in an orthogroup and the mapping ratio (MR) is not often cited in the correlation analyses they perform. They typically cite CR, which is capturing proportion of identical residues in the mapped residues. For orthologs that retain the same function I would expect many residues to superpose well (as they typically find) so the CR ratio would reflect changes in physico-chemical properties to some extent. However, the CR ratio does not consider the degree of physico-chemical change e.g. they don't use any metrics to measure the degree of change in physicochemical properties. That could be interesting to consider in the future.

So the term is not going to be that different from sequence identity except for orthologs that don't superpose well. However, if I have understood correctly, for these remote orthologs that don't map well actual structural divergence is lost in the CR ratio as they normalize by the number of mapped residues rather than the length of the reference structure. I'm not sure why that method was chosen as it doesn't exploit the structural information from AlphaFold as fully as it might.

The CR value they use will miss some structural divergence e.g. inserted residues in the region of active site loops that may enable binding to a different substrate. The information on these unmappable inserted residues would be lost by considering CR on its own. Perhaps the authors can provide more explanation for this choice of metric.

Furthermore, for this reason, it might be good to use a term like 'divergence' rather than 'structural divergence' in the various studies or at least to discuss the caveats of the term they use, as they don't seem to be fully measuring structural divergence, unless I have misunderstood.

4. How the orthogroups are generated is clear as they are generated using OrthoMCL and refined by structure comparison, but there seems to be subsequent clustering based on carbon source or species, which I found rather confusing. The authors describe these as subgroups of the orthogroups, which I assume means sets of orthogroups or are they subsets within orthogroups? It was a bit confusing and it may be helpful to rephrase or have an example in supplementary to illustrate this better.

5. The observation that enzymes involved in anaerobic fermentation are more constrained is very interesting but I was not sure I completely understood why this might be the case and would have appreciated more discussion. Is it because

anaerobic fermentation is the more cost-effective process and residue changes could impact detrimentally on that?

6. Some other observations are very easy to grasp such as observations that orthologous groups with more protein interactors or involved in large complexes were more likely to be constrained. Furthermore, it is also intuitive that critical processes like central carbon metabolism and highly constrained enzymes like oxido-reductases have high CR. For enzymes like the hydrolases could the low CR reflect residue changes that are optimising the chemistry or modifying specificity to better exploit nutrient opportunities? I felt that tuning of the chemical reaction was hinted at in the section on changes in flux but it was a bit hard to follow that thread between sections.

7. Another observation that I found hard to intuitively grasp was that enzymes with highly conserved surfaces tend to be less costly. Could the authors suggest some reasons for this? Could this be because these enzymes are involved in complexes with high abundance, and it is the abundance that relates to the low cost? Perhaps I missed some details but there are many aspects to hold in the mind, which makes it hard to follow sometimes.

8. It is interesting that the authors find that a significant proportion of the conserved residue clusters do not overlap with binding sites. It would be good to have a bit more clarity on this. Not all proteins have direct experimental information on binding sites. It would be helpful to know how many of the proteins and orthogroups in their dataset did have experimental functional sites data extracted from PDB structures. It is also possible that some of the ligands in PDB structures are not cognate ligands - so some cognate binding residues could be missing. These clusters that don't overlap known binding sites could also reflect residues involved in allostery, which are not likely to be comprehensively annotated in PDB or UniProt. The authors may want to discuss these issues.

Furthermore, I wasn't sure whether the authors were suggesting that the clusters were missing some of the characterised binding sites, which would be a rather surprising observation. Perhaps that section could be phrased a bit more clearly.

9. As regards residue preferences, could the correlation regarding glycine content and conservation in random coil reflect the fact that glycine is often used for turns in the protein fold, which would tend to occur in residues at the surface of the protein not allocated to secondary structures? Do the authors see any trends for conserved glycine residues being associated with tight turns for example?

In addition, as the authors point out they are unlikely to have surveyed random coil portions comprehensively as these are quite likely to be unmapped for many relatives in an orthogroup. The residues that map are indeed perhaps more likely to be involved in conserved regions valuable for the conformation.

10. The observation of sites under positive selection locating at protein interfaces in structurally divergent orthogroups is another interesting observation and a phenomenon that seems reasonable given the fitness benefits from the enhanced stability of the oligomers. I was quite surprised there was not a signal for other types of functional sites though. Perhaps this suggests that most orthogroups are in fact associated with functionally identical relatives.

11. Since the orthogroups will belong to larger evolutionary superfamilies having particular structural characteristics, I was curious to know whether the specific structural arrangements that these superfamilies adopt in their core could also aid or constrain structural divergence. It may be interesting to map the orthogroups to superfamilies in ECOD or SCOP and determine whether some superfamilies are more enriched in structurally divergent orthogroups. However, the article is already very rich in detail and that might be something for a future study.

I found this article extremely interesting but it is incredibly dense. I read it multiple times but still wasn't sure if I fully appreciated the meaning of all the findings. For example, to what extent does diversity reflect specific benefits? For example, do residue changes alter enzyme specificity in order to adapt to different environments? Or do residue changes tune the chemistry and the $K_{cat}/flux$, or do residue changes optimize protein-protein interactions, thereby stabilizing complexes? I think the authors are suggesting that residue changes could be contributing to optimization of multiple properties, and the balance depends on the particular enzyme and pathway involved. However, the study covers so much that at times it is quite hard to keep track of the key fundamentals amongst the complexity. Nevertheless, I found it exciting and deeply interesting.

Referee #3

(Remarks to the Author)

This study explores how metabolism and structural evolution influence enzyme evolution over 400 million years. The findings reveal that:

1. Overall Metabolism: Metabolic processes shape enzyme structures, with evolution linked to reaction mechanisms and metabolic flux variability.
2. Metabolic Network Organization: The structure of the metabolic network influences enzyme evolution, affecting how enzymes interact within the network.
3. Enzymes' Molecular Properties: Evolutionary optimization varies across enzyme domains, prioritizing small-molecule binding sites, with surface residues adapting for protein-protein interactions.

Overall, I find the paper highly original and significant, with a clear, well-written manuscript that effectively links different disciplines. The findings are novel and offer valuable insights into how evolutionary constraints impact protein structure within a cellular (metabolic) environment. The study is robust in its data and conclusions, and it holds great relevance for protein scientists, molecular evolutionists, and the microbiology/synthetic biology community.

However, I have three major concerns/comments:

1. In the paragraph starting at line 259 "in parallel, we perform Go-slim..." the authors concluded that the metabolic specialization drives the structural evolution of enzymes that belong to these metabolic pathways. However, they never discuss the null hypothesis: what about if it is the other way around? What about if the enzyme structure shapes the metabolism? In principle first we have the emergence of enzymes and then they organise themselves in metabolic pathways.
2. My second concern follows the previous comment: if any of the species has any specialised pathways not common to others, is it possible to explore these ones? And detect the differences? This data would add robustness to their results and discussion.
3. To strengthen the manuscript, I suggest incorporating an analysis of fold usage, given the structural analysis already performed. It would be evolutionarily interesting to see if there is a bias in fold usage and whether the most catalytically diverse folds, such as the TIM barrel and Rossmann fold, appear in this context. Including this analysis would add another layer of depth to the study.

Other minor comments/corrections:

Abstract line 47: "on the variability of flux (?) rather than..."

Intro line 62: "enzyme plasticity is important for across disciplines"

Results line 173: The authors mention 7 major enzyme classes, the last one is "2 translocases". These translocases are not addressed anywhere else in the paper; is there a reason? If so, can it be added in the text? If not, can this class be discussed as well (e.g. Fig. 3c includes only the first six classes of enzymes)?

Line 423, "We calculated the average cost per amino acid for each protein averaged over the orthogroup using different cost metrics": Does this mean that the authors determine the average cost per amino acid for each yeast species? Is this consistent across all species?

Version 1:

Reviewer comments:

Referee #1

(Remarks to the Author)

The authors responded reasonably to my concerns from the first round of review. They performed several additional requested computational analyses. My one major remaining concern, which I acknowledge is a challenge to address, relates to the lack of experimental validation of a meaningful (but not onerous) subset of computational predictions. There was essentially none of this performed.

Additional points:

Regarding the statement in the response: "Furthermore, oxidoreductases are indeed more conserved overall than other enzymes, however, this is not solely due to their dependency on a cofactor: this relationship only holds true as long as one includes the oxidoreductases of central and high-flux pathways - glycolysis, gluconeogenesis and glutamate biosynthesis - that are rich in oxidoreductases. Once one excludes the oxidoreductases participating in these pathways, the oxidoreductases are not more conserved than other enzyme classes."

If this is true, why not split out central and non-central oxidoreductases to highlight this relationship in the text? If this is there, I missed it.

- Ln 447-448: Is the appropriate figure being referenced here? I don't see data for the loop region prediction error related question.

- Ln 493: Please define the ML model, what specifics?

- Several of the figures seem to be missing colormaps (e.g. Fig. 3c) or appropriate definitions (e.g. "1" vs "0" in Fig. S2.1a) and descriptions (e.g. groups being compared for enrichment in S2.2).

Referee #2

(Remarks to the Author)

I have checked the authors responses and revisions to the paper. I'm grateful for the detailed answers and the additional

research that the authors performed in response to my questions and suggestions. I find the revised manuscript much clearer.

It's helpful to know that most orthogroup relatives share the same EC4 number. Apologies if I missed that detail before. It's useful to have the information on levels of experimental annotation.

I thank the reviewers for their analysis of fold groups in the TED classification and found it interesting and maybe not surprising that the Rossman and TIM barrel fold groups were slightly enriched in the most divergent orthogroups. These fold groups are likely to be enriched in enzyme superfamilies which may have optimised the mechanism or efficiency of their reactions.

One minor comment remains which does not need to be implemented. It was helpful to have more explanation for the choice of TMscore. It might be helpful to remind readers of the equation for this score as that would make it clearer that it better captures global similarity than RMSD.

I thank the authors again for a fascinating and rigorous article.

(Remarks on code availability)

I have not been able to review the code.

Referee #3

(Remarks to the Author)

The authors have made substantial improvements to the manuscript by thoughtfully addressing the concerns raised in my previous review. The additional explanations and clarifications provided significantly enhance the context for their observations and strengthen the overall observation.

We thank all reviewers for their time and the productive feedback they provided, which helped us to improve our manuscript. We have addressed their suggestions and concerns in the following way:

Referee #1:

Lemke and Heineke et al. leverage Alphafold2 and a series of evolutionary and computational analyses to investigate the structural evolution of metabolic enzymes using 27 Saccharomycotina species. They propose that instead of structure dictating function, instead metabolism has shaped enzyme evolution via the topological organization of the metabolic network and the enzyme's molecular properties. While a compelling hypothesis is put forward, the authors never provide clear causal evidence that metabolism is driving the evolution of enzyme structure and not the inverse.

While both scenarios are plausible, there are several alternate hypotheses to the ones the authors claim that are not sufficiently investigated. Additionally, the claims the authors make are not experimentally validated, or otherwise lack additional in silico experiments that are not already predicated on previous predictive models. We therefore suggest that this manuscript, while potentially impactful, is not acceptable for publication in its current form. While the study addresses an interesting topic, several significant issues must be addressed before publication. Moreover, there is some concern that, to rise to full impact, the manuscript would require experimental validation that is not feasible in a reasonable timeframe.

We thank the reviewers for their valuable time and effort in assessing our manuscript. In response to their feedback, we have addressed their specific comments (as detailed in the point-by-point response) and have reworked the discussion to put more emphasis on alternative hypotheses - several which are not mutually exclusive (please see the reply to comment #1 of Reviewer #3 on pages 24-25 in this document, and the first paragraphs of the revised discussion, on page 20 of the revised main manuscript). We acknowledge that the large-scale nature of our study—analyzing over 11,000 enzyme structures across 27 different species—limits the extent to which the general principles uncovered can be validated through experiments done on individual enzyme structures. However, understanding the reviewer's desire to have more experimental results to back our findings, we have performed several new analyses on a large set of experimentally determined data, including protein structures obtained from PDB, enzymatic data extracted from UniProt, and experimentally determined fluxes. Please see the individual replies below, and the text updates in lines 184-186 and 253-254 in main section 2 of the results, lines 316-317 in main section 3 of the results, lines 514-545 and 634-639 in discussion, Supplementary Note 2.

Fig. 1:

Many of the enzymes analyzed in this study operate as multimers. However, vanilla AlphaFold v2 (from the 2021 publication) is not tuned to predict multimeric structures. Did the authors investigate a combined approach using AlphaFold for monomers and AlphaFold-Multimer for multimeric enzymes? There should be a more thorough discussion of the biases between these methods, including those that arise in combining predicted structures from the two. Some of the variations discussed in enzyme classes could have resulted from poor predictions for enzymes that assemble as multimers, since base AlphaFold does a poor job of predicting a multimeric structure.

We thank the reviewer for this comment. To address whether the frequent participation of enzymes in complexes might bias our analyses due to the use of Alphafold2, we used experimentally determined structures to test whether they are equally well predicted. We obtained 553 experimentally determined homo- and heteromeric complex structures from RCSB PDB based on 106 reference structures. In addition, we obtained 108 experimentally determined monomeric structures based on 34 reference structures. Then, we compared these experimentally determined structures with the ones predicted by Alphafold2. Comparing both data sets, we observe no significant difference between enzymes present in multimers and those which are not (Response Figure R1 (left), now Supplementary Figure S5.2C in the manuscript). To assess this also in another way, we compared the average mapping ratio for orthogroups that were identified as within complexes in ComplexeomeDB against those that were not. Again, we detect no significant difference (Response Figure R1 (right), now Supplementary Figure S.5.2D in the manuscript). We added this analysis to the discussion (lines 551-555) and to the Supplementary Note 2 (lines 703-716).

Response Figure R1, now Figure S5.2C,D in the main test Left: Predicted *S.cer.* structure aligned to corresponding crystal structure. The mean mapping ratio per reference structure is shown. Right: Comparison of proteins in complexes (ComplexomeDB) and proteins that are not in complexes. Statistical testing was performed with a two-sided Wilcoxon-Mann-Whitney test.

We also tested the reviewer's concern that different propensity to form multimers might exist between different enzyme classes leading to a bias, for instance, for hydrolases. Even though hydrolases are the most divergent enzyme class when considering only mapped residues (Conservation Ratio), the Mapping Ratio for hydrolases (84.9%) was only slightly smaller and not statistically significantly different than the mapping ratio for all enzymes (87.4%) (New Figure S6.1F). Indeed, hydrolase conservation has a lower correlation with mapping ratio (Pearson $r = 0.29$) than on average (Pearson $r = 0.48$) (New Figure S6.1A). Thus, the different relationships between conservation and other properties we detect for hydrolases relative to other enzyme classes are not based on a poor mapping ratio.

In order to address the Reviewer's question, we have included this information in our manuscript, alongside an expanded discussion of hydrolases (see the comment to Figure 2c below, pages 5-6 and lines 616-622 in the manuscript)

Finally, the reviewer asked us to comment on AlphaFold multimer, which is a highly useful AlphaFold-extension for predicting the relative position of protein subunits in multimeric protein complexes, on the basis of AlphaFold2 predicted protein structures, a property that we have not studied herein. The tool is cited in lines 705-707 and as suggested by the reviewer we have added a specific discussion point to Supplementary Note 2.

Fig. 2:

2a: The authors claim that enzymes from fermenting species showed higher structural conservation relative to *S. cerevisiae*. This claim is interesting only if it can be distinguished that this is not merely an effect of evolutionary proximity. For instance, the sampled species from the phylogenetic tree do not appear to be randomly distributed in evolutionary time. It is thus not clear if this apparent structural conservation is just a result of sampling bias. Are fermenting species closer in evolutionary time than they are to respiring species? The authors should also comment on the functional definition of fermentative vs. non-fermentative species as this makes a difference in the interpretation. Are the species labeled non-fermentative unable to ferment, or do they simply prefer not to? Many species can do both but have a strong preference for one. This is another potential bias that could change how these results are interpreted.

We apologise that our wording of this section was unclear. Glucose fermenting species are, with some exceptions, evolutionary closer to each other than non-fermenting species within the *Saccharomycotina* subphylum (illustrated in new Supplementary Figure S2.1A, based on data from (Kurtzman et al., 2011; Lu et al., 2021)). However, our analysis takes this into account, as we focused on the largest differences in conservation between subsets of enzymes from yeasts with or without the glucose fermentation phenotype. The interesting observation we made is that some of the strongest differences observed within this group occurred in orthogroups of enzymes specifically implicated in central carbon metabolism. Conservation for these enzymes differs much more than the difference expected by phylogenetic distance (Figure 2A, new Supplementary Figure S2.1B). For example, Kgd2p which catalyzes the oxidative decarboxylation of alpha-ketoglutarate to succinyl-CoA in the TCA cycle is much more conserved in fermenting species relative to non-fermenting species than would be expected based on phylogeny alone, while Ald5p, the mitochondrial aldehyde dehydrogenase which helps to regulate biosynthesis of electron transport chain components is more conserved in non-fermentative species.

In order to address the reviewer's point and to avoid any misconception, we have revised the paragraph intensively. The text now reads (lines 183-207):

"We asked whether metabolic specialisations at the species level are reflected in the protein structures. For this analysis, we used growth properties of the yeasts on 21 different carbon sources (Kurtzman et al., 2011; Lu et al., 2021; Opulente et al., 2018; Shen et al., 2018)(Figure S2.1A). Enzyme structures that depend on whether a species can ferment glucose, raffinose, galactose and sucrose exhibited the most significant differences in average CR between subgroups, alongside enzymes from species that grew aerobically on D-xylose (adj. p-values < 1e-83, two-sided Wilcoxon signed-rank test). For enzyme structures from anaerobically fermenting species, we revealed a higher conservation relative to the structure from *S. cerevisiae*, which also ferments those sugars (Figure 2A, S2.1B). While this finding corresponds to their closer phylogenetic relationship, some of the largest differences in conservation between the subgroups were detected within the orthogroups of enzymes which

are directly involved in the processes, including central carbon metabolism and the electron transport chain (ETC). For example, Kgd2p of the TCA cycle and Cox7p of the respiratory chain, were among the enzymes exhibiting a higher-than-expected change in diversity between the fermenting and non-fermenting species. These more divergent orthogroups also included Met10p of the methionine and sulphur cycle; Ath1p of trehalose metabolism, which in yeast serves as carbon storage; and Erg1p, involved in the biosynthesis of ergosterol. We also observed a few cases in which the CR in non-fermenting species was higher than in the fermenting species. Also in this case, the enzymes were directly related to oxidative metabolism, including Ndi1p, an NADH dehydrogenase which is a functional analogue of the respiratory chain complex I; Ald5p, an aldehyde dehydrogenase; Idp1p, an isocitrate dehydrogenase; and Ilv6p of branched-chain amino acid metabolism. Moreover, GO-slim terms “membrane”, “lipid metabolism”, “endoplasmic reticulum” and “endomembrane system” were enriched in the first quartile of the orthogroups with the largest differences in CR between subgroups for glucose fermentation (adj. $p < 1e-2$, Fisher’s exact test) (Figure S2.1C).”

Furthermore, to address the request of the reviewer, we have added the applied definition of fermentation, which follows the classification by (Kurtzman et al., 2011, pgs. 100-101). Therein, a yeast species is defined as fermenting if they produce Carbon Dioxide over a period of up to 28 days at 25-28 °C using inverted Durham tubes. We have adjusted the text in the manuscript, methods section, and figure caption to more clearly indicate the source and nature of the data.

Of note, the section is further strengthened in the revision by an additional analysis that we have conducted in response to Reviewer #3, comment #2, in which we have included a similar analysis of the Xylose pathway. This is a more specialized pathway and should further illustrate how metabolic specialisation at the species level can help drive enzyme structural evolution (Lines 209-219).

2c: One of the conclusions made from this figure is that hydrolases, which have more disordered regions, are less conserved in structure. However, there may be a trivial explanation here: hydrolases have a dramatically more straightforward and structurally localized mechanism, whereas oxidoreductases require binding of both a substrate and a large cofactor, necessitating a more complex structural solution, which is thus more likely to be evolutionarily constrained. If this is true, then there is seemingly little to be learned from these data. To be able to make the claim the authors wish to make, there are several alternative hypotheses that could be investigated.

We thank the reviewer for their comments. A key finding of our study is indeed that the reaction mechanism and metabolic function of an enzyme is a key constraint for its evolution across the structural levels. Our data, though, shows that the relationship is rather complex, depending on multiple factors, and this holds true for hydrolases. To illustrate this point better, we have specifically contrasted the structural diversity of the two enzyme classes the Reviewers highlight, hydrolases (least constrained reaction mechanism) and oxidoreductases (highly constrained reaction mechanism). Hydrolases, despite being more diverse within mapped

regions, do not have a significantly different mapping ratio compared to other enzymes (New Figure S6.1F), and they also do not have a significant difference in mean pLDDT which is consistent that they don't have a higher content of intrinsically unstructured regions. We apologize if our presentation of results led to a different interpretation, which was not intended. We have now included a relevant analysis and discussion point (Lines 622-630).

Furthermore, oxidoreductases are indeed more conserved overall than other enzymes, however, this is not solely due to their dependency on a cofactor: this relationship only holds true as long as one includes the oxidoreductases of central and high-flux pathways - glycolysis, gluconeogenesis and glutamate biosynthesis - that are rich in oxidoreductases. Once one excludes the oxidoreductases participating in these pathways, the oxidoreductases are not more conserved than other enzyme classes. In contrast, the diversity in hydrolases is not dependent on specific pathways or GO-terms enriched for more divergent orthogroups (such lipid metabolism, that contains 6 of 55 hydrolases). Hence, while our results show that the constrained reaction mechanism contributes to enzyme structural evolution, other key factors (we demonstrate the participation in central or high flux pathways, flux, abundance, and small molecule interactions) can trump this relationship. In order to address the reviewer's question, we have emphasized this point (lines 259-265), added a paragraph to the discussion, that specifically addresses the specific features of hydrolases including new results as Supplementary Figure S6.1 (lines 615-630). See also our response to Reviewer #2 comment #6, page 18.

Next to 2f: Given a large enough sample size, small differences in distributions can lead to a "significant" p-value. Although the Wilcoxon-Mann-Whitney test indicates a significant difference, the data exhibit only a slight shift between groups, the quartile ranges are similar, and the median lines lie within the overlap of the quartile ranges, indicating no obvious difference in medians. The authors should include a calculation of effect size or other appropriate metric to determine if these differences are likely to be biologically meaningful.

We agree with the Reviewer that statistical significance alone can be misleading to judge the relevance of a result. To address the reviewer's comment, we now include measures of the effect size for this result. The difference in median conservation ratio is 0.05 (7.6% decrease in conservation). Using Cliff's Delta as a measure of effect size results in a value of 0.23, corresponding to a small to medium effect size. We would like to highlight that the functions of enzymes that interact with metal ions are highly heterogeneous (metal ions are needed across the metabolic network in a broad and various set of different functions (Aulakh et al., 2024), thus reaching a medium effect size on average, is within expectation for a significant result. We have added this information to the main text, plus a caveat highlighting the highly heterogeneous nature of the metal binding information (lines 268-271).

Fig. 3:

3d: We are concerned with the authors' claims of predicting flux. Accurately estimating flux across species is likely to be highly inaccurate, biased by factors such as substrate availability, protein conformation, and more beyond the simple issues of protein steady-state abundance. The authors need to provide reasons to be confident that these models are accurately reflecting flux through these reactions. At a minimum, we might expect a simple glucose tracing experiment between strains measuring experimental flux through glycolysis or the TCA cycle compared to the flux predictions made using steady-state protein values from these pathways to determine the general accuracy of these flux predictions.

To verify whether flux estimated using genome scale models resembles flux data obtained from ^{13}C labelling experiments, we compared the median simulated fluxes through orthogroups of central pathways with ^{13}C fluxes from 6 studies that focused on central metabolism (Blank et al., 2005; Daran-Lapujade et al., 2004; Gombert et al., 2001; Jouhten et al., 2008; Kajihata et al., 2015; Raghevendran et al., 2004). The Pearson correlation coefficient for these correlations is approximately 0.8 when we include reactions with at least 5 studies reporting measured ^{13}C fluxes. The experimental flux measurements, which cover only central metabolism, are thus in good agreement with the genome-scale models that we used to estimate the broader, network wide fluxes. We include this analysis as Supplementary Figure S3.1C, lines 315-317 and lines 1325-1328.

Fig. 4:

The concept of "cost" is quite important to the interpretations of the manuscript, but the concept seems quite opaque and somewhat arbitrary. More needs to be done to educate the reader on what is meant by "cost" and to convince them that it is being estimated appropriately. We find it hard to imagine that the cost of an amino acid is the same for all species, even within a defined subphylum, especially when they live very different lifestyles. If the cost for amino acids were different between species this would compromise the analysis.

We apologize that our previous version included no in-depth description of the concept of cost and cost calculations. We had chosen this strategy simply due to limited space - as we had not introduced these metrics or their concept as part of our paper; we had referred to the original papers instead. In order to be more informative for the reader, we now created a Supplementary Note which reviews the eight applied cost metrics (Supplementary Note 4, lines 769-800). In brief, the commonly used cost estimates range from simple cost assessments that are based on atomic composition molecular weight (Seligman 2003), to highly sophisticated network-based matrices, that calculate synthesis costs based on the individual genome scale reconstructions of metabolic network (Chen and Nielsen 2022).

The reviewer further makes an interesting point, and asks whether species specific differences in metabolic networks, and resultant differences in synthesis costs could confound our results. In order to test for this possibility, we have compared network-based cost metrics with those that are independent of the species-specific metabolic networks (i.e. those which are based on molecular weight or energy equivalents within core metabolic pathways common to all species), with the metrics that are based on species specific metabolic networks. We find that both types of cost metrics point to the same cost optimization principles (Figure 4A,B in the manuscript). We can thus conclude that our results are robust to divergence of the metabolic network structure, and species-specific differences in cost. We have added this point to the methods section (lines 1268-1278) and to a Supplementary Note 4.

Fig. 5:

5b: If a sub-domain could be either an alpha-helix or beta-sheet, it has been proposed that AlphaFold might have a bias to predict a helix preferentially over a sheet (<https://www.ncbi.nlm.nih.gov/pmc/articles/PMC9312937/>), so of course helices will end up being more variable. The authors should assess and address this potential concern with the AlphaFold model.

We thank the reviewer for pointing out this potential bias of Alpha Fold2. The result from (Stevens and He 2022) suggests that in loop regions of proteins Alphafold2 incorrectly predicts the presence of alpha helices at a higher rate (2.02% of loop residues) than it does for beta sheets (0.26% of loop residues). In order to test if this bias affects our conclusions, we calculated the expected error on the basis of the frequency of loop occurrence in our data (40.8 %) expecting 2.28% wrong classification by AlphaFold2, as a conservative estimate of the error. This results in an estimated error of 1.0% compared to a mean difference (MAE) of 9.7% for the comparison of helical versus extended conservation. Our result is robust when considering this error margin. In response to the Reviewer's question, we have added our analysis to the manuscript (lines 444-448) and (lines 553-554).

5e: In our opinion, the most impactful implication of this manuscript is the ability to predict a priori small molecule binding sites. While the authors show one interesting example here, this is not sufficient validation or meaningful follow-up of these claims, which seem to be a central claim of this work. More could be done to expand this meaningful outcome of the work.

We agree with this suggestion and in our revised manuscript we extend this line of investigation in two directions.

- 1) We implemented a machine learning workflow based on binding site data extracted from UniProt for 302 reference structures and 241 orthogroups, to estimate how well binding sites are predictable from the

presence of a cluster of full structural conservation. We included amino acid content and physicochemical properties as features. Using 10 runs of 5-fold cross-validation, our predictions show a good performance (Average Balanced accuracy of 0.63, average AUC of 0.68 on test data that were not included in the training procedure) in classifying binding sites. For context, we compared the predictor with two “randomized” sets, either randomizing the binding site residues (which results in a stronger class imbalance and has thus to be analyzed carefully) or randomizing the labels (which conserves the class balance but adds a bias for similar clusters). In both cases the predictor outperforms the random models (Balanced accuracy of 0.57, AUC of 0.60 (random binding site) and balanced accuracy 0.53, AUC of 0.54 (randomized labels)).

Also, projecting the prediction probability onto individual amino acids confirms that residues in known binding sites are more likely to be predicted as part of binding sites using our predictor than residues outside of known binding sites (p-value < 1e-38, two-sided Wilcoxon signed-rank test, adj. Cliff’s delta = 0.78). Since the binding site annotations are incomplete, we have an expected error in our model, however, this analysis shows that amino acids with a higher predicted probability are more likely to be part of a small molecule binding site. These results (Response Figure R2) have been incorporated in the main manuscript (New Supplementary Figure S5.1G,H, lines 493-500).

Response Figure R2, now Figure S5.1G,H: Performance of a trained machine learning model to predict known binding sites (left) compared to two random models: randomized binding site indices (center) and randomized labels (right)

- 2) As not all clusters map to a small molecule binding site, we tested whether highly conserved regions could also overlap with protein-protein coordination sites, as they do with small-molecule binding sites, a point highlighted by Reviewer #2 (comment #8, page 19). To assess this, we gathered coordination sites for 106 of our reference structures from 546 experimentally determined structures in PDB. For orthogroups

containing a structure with a known coordination site, the distribution of CR is higher for coordination sites than for full proteins, and we see a similar result between the surface residues of the coordination site and general surface residues (new Supplementary Figure S5.1J). For a fair comparison, for this analysis, we only took into account orthogroups for which we could extract a coordination site.

The highly conserved clusters of residues that we identify within each orthogroup are slightly overrepresented in residues that belong to these coordination sites (new Supplementary Figure S5I), just as they are for binding sites (Supplementary Figure 5G), although the effect is weaker (adjusted Cliff's Delta of 0.23 for coordination sites vs. 0.84 (CSb) and 0.92 (UPb) for binding sites). These results have been incorporated in the manuscript (lines 500-504).

Fig. 6:

It's not obvious from this figure or the text why there would be positive selection at protein interfaces. Would this be to increase or decrease affinity of the interaction? Perhaps, it could change the properties of the complex in some advantageous manner? The authors should help us understand the impact of this section of the manuscript. Again, this would require either experimental validation, or validation using additional data and not just predictions already predicated upon a prediction (e.g. the AlphaFold structure).

We thank the Reviewer for this comment. We believe this section adds value to our study, because it demonstrates that a combination of structural biology and functional genomics can provide deeper understanding of structural elements under positive selection. We agree, however, that this is not a central point, and as part of the general suggestion by all Reviewers to shorten and simplify our paper, we have shortened this section significantly in the main text and put less emphasis on it in the revised version.

Furthermore, in order to test if there could be a common mechanism that explains the positive selection of the identified residues, we have expanded our search for corresponding structural elements. This search identified several residues, and these are located in different structural elements. We highlight these in a revised Supplementary Figure S5.4 in three examples. In the enzyme Pox1p, one positively selected site is in the dimeric protein-protein-interaction site, in the respiratory chain proteins we observe positive selection on several complex interaction interfaces. Our search also identified positively selected residues in glycolytic enzyme GAPDH, where they are located not at the protein-protein interaction sites of the tetramer, but primarily in surface-exposed residues. Here, we also detect one positively selected amino acid which is in proximity of the catalytic site. Our expanded analysis thus implies that a multitude of biological mechanisms can result in the positive selection

of a specific structural element. We have thus revised this part of the manuscript and associated text (lines 462-478). We agree that identifying the biological reasons for the positive selection would require follow up studies. However, because the biological reasons are different for each positively selected residue (i.e. we cannot generalize from one or a few examples for the others), we believe that such studies are out of scope of the present manuscript. We have thus expanded Supplementary Note 6, discussing that the impact of our study in this respect is that it identifies the positively selected residues in structural elements, as a hypothesis generating starting point for future studies.

As an additional note, some of the explanations are at times overly complicated and difficult to follow. We acknowledge that this paper is computationally intensive and a sophisticated topic, but the authors could improve the accessibility. We appreciate that the analysis code and the code for the figures are shared. If the authors intend this work to be a reusable resource for bioengineers, as mentioned multiple times throughout the manuscript, we would recommend they consider publishing the code as interactive Jupyter notebooks with detailed documentation. This will enhance its usability and enable effective reuse.

We thank the reviewer for this feedback. We have worked extensively on the text and figures to make them more accessible.

As suggested, we have provided a Jupyter notebook, in the corresponding GitHub repository: https://github.com/OliverLemke/structure_comparison/tree/main/Example. This repository contains a Markdown file with a step-by-step description and examples for the mapping as well as the calculation of certain properties that do not require external resources. We mention it in the Code Availability statement (lines 1498-1499)

Referee #2:

The article is very interesting with many novel observations. It describes a comprehensive and robust study of how metabolic processes have driven the evolution of enzyme structures over the last 400 million years. It is dense in interesting observations and yet overall mostly written in an accessible style with key findings summarized in the discussion. It is extremely timely benefitting from high quality AlphaFold structures that allow identification of structurally equivalent positions between relatives which are subsequently analysed for conservation of amino acid residue. This mapping enabled a large scale study correlating residue changes in enzymes with differences in various metabolic features. It also allowed the authors to analyse the evolutionary constraints acting on different regions in the enzyme structure. The study is impressive in its broad scope, analysing at different scales from pathways to enzymes and by exploiting the structural data, to enzyme

substructures.

The authors selected a sound set of yeast fusion species from the Saccharomycotina subphylum for the study associated with a significant amount of experimental data. This allowed them to correlate what they describe as ‘structural divergence’ of enzymes within orthogroups with a wide range of interesting features including growth properties, protein abundance, variation in metabolic flux, sites under positive selection etc. The authors have used robust statistics throughout and highlight statistically significant observations.

They employ a careful selection of orthogroups ensuring these contain genes that cover a substantial proportion (40%) of the species analysed. They were able to generate a substantial number of high quality AlphaFold structures for the study, and it is particularly commendable that they exploited this structural data to refine the orthogroups.

Evolutionary variation is quantified by metrics that capture the mapping between enzymes and the proportion of mapped residues (superimposed on the reference structure within 2Å) that are identical. They use this conservation ratio as a proxy for structural divergence. It is measured against the reference *S. cerevisiae* structure in each orthologous group. I am not sure that ‘structural divergence’ is the most appropriate description of what they are analysing, but I discuss this more below.

Many of the trends the authors observe are not surprising and easy to grasp intuitively i.e. that enzyme sites implicated in function (e.g. interactions with substrates or proteins) are more highly conserved. However, there is a lot of fascinating detail throughout and a rather complex picture emerges involving a balance between multiple aspects (abundance, enzyme cost, functional constraints). I sometimes found it hard to keep track of the narrative as so many aspects are being considered. The figures are very good and helped a lot as did the discussion at the end, but I wondered if there were additional strategies that could help the reader keep track of the key findings as they emerge.

In summary, this is an extremely interesting and valuable study. A particularly useful aspect of the study is that it provides a comprehensive analytical framework that will be extremely valuable for other research groups studying or optimizing the properties of specific enzymes.

Minor Comments

1. The refinement of orthogroups using structure comparison and clustering was valuable given the correlation between structural similarity and functional similarity. It would be good to have a clearer understanding of the threshold used for this refined clustering, and how this had been decided. For example, do domains within the

revised orthogroups cluster within 2 Angstroms? They mention the tool they used but it would be good if they cited the threshold in the main text and gave a widely used measure like RMSD, if possible.

We thank the reviewer for their input. In order to address their question, we have added details to our results and our methods section (lines 133-135, 1052-1055) to give a better overview of the thresholding strategies used to refine our orthogroups. In brief, our final list of 429 refined orthogroups were defined based on a hierarchical clustering after computing TM-align scores between clusters. Clusters were defined based on a linkage cutoff of 0.2 chosen for two reasons. This threshold improved the average TM-score for our orthogroups (0.773 after clustering vs. 0.710 before for the 29 subclustered orthogroups), indicating better structural alignments. Secondly, it maintained clusters with a sufficient spread of proteins from different species in at least one subcluster from each orthogroup. We reported the TM-scores of the split orthogroups in Supplementary Table S1.5. We also include a new Supplementary Table S1.6 which includes TM-scores for all orthogroups whether or not they were split into subclusters.

We decided against the use of RMSD since RMSD needs information from residues with pairwise 1:1 mapping, which can be helpful in analyzing conformational changes or smaller mutations. As we were also interested in global fold similarity, we used the more versatile TM-score.

Are all relatives in the final clusters likely to have the same enzyme chemistry and specificity or could there be divergence in these properties e.g. in specificity? It is known that more remotely related orthologs can diverge in function. Was there some evaluation of clustering threshold value based on structural similarity between enzymes with identical Enzyme Classifications at the 4th level (EC4), supporting identical chemistry and substrate specificity? The clustering threshold is key to understanding whether the low CR of an orthogroup reflects a functionally identical set of relatives (same EC4) where residue changes are tuning functional or stability properties or a group of relatives in which there has also been some change in functional property e.g. change in the substrate acted upon.

We thank the reviewer for their comment. Considering the subset of enzymes with direct experimental evidence of the main enzymatic function (based on a PubMed evidence code), there were no cases where the enzyme function changed at EC levels 1, 2, or 3 within the orthogroups. Furthermore, there were only five cases, where there are shifts between chemically similar reactions (EC-level 4). For example, OG1390 contains three paralogous proteins in *S. cerevisiae*, Hsu1p, Str2p, and YML082Wp. Str2p and Hsu1p catalyse closely related but not identical enzymatic reactions; Str2p is a Cystathionine Gamma Synthase (EC 2.5.1.48), and Hsu1p a Homocysteine Synthase (EC2.5.1.49), and Cysteine Synthase (EC 2.5.1.47)). Notably, the Homocysteine Synthase activity of *Hsu1p* is of very low rate; it is thus worth speculating that this activity derives from a promiscuous reaction (Oss

et al., 2022; Sonal et al., 2023; Yu et al., 2022). We also tested whether these 5 orthogroups have a significant change in conservation compared to the other orthogroups, however there was not a significant signal. Based on available data, as accrued in UniProt, we can thus conclude that, overall, enzymes within a structural orthogroup will typically catalyze identical or very similar biochemical reactions. This assumption is also consistent with the assumptions of genome-scale network reconstructions of these species which have been used successfully to describe metabolic flux and growth on different substrates (Lu et al., 2021). We have updated the manuscript to include this information (lines 252-257, Supplementary Note 3.2), and emphasised this point better in the discussion (lines 604-610).

2. Following on from that, to what extent are the relatives annotated? Since the authors examine whether specific types of enzymes (oxidoreductases, hydrolases) are less/more structurally divergent, I assume they have used experimental data from the Enzyme Classification or predicted annotations from UniProt to identify these types of enzymes. What proportion of the assignments are experimental? Do they see any variation in the functional annotations across orthogroups? Apologies if I missed a description of this in the supplementary. It would be helpful to have some mention in the main text, too.

For our assignment of enzyme classification, we use the information collected in UniProt (The UniProt Consortium, 2025). In order to independently assess the amount of primary experimental information on which Uniprot function assignment is based, we searched the 529 reference structures. For 45% of these we identified at least one PubMed evidence code assigned for the enzyme class annotation. In addition, for the other 26 species, 675 structures have at least one PubMed evidence code assigned. Looking at the data from a different perspective, for 468 of our reference structures EC numbers are assigned in Uniprot, of which 238 (51%) have direct Pubmed evidence codes. The function of the remaining enzymes is inferred from homology by Uniprot. We added this information to the results and the methods section (Lines 243-245, Table 1). Please see our reply to the previous comment about the variation in EC annotation within an orthogroup; lines 252-257.

3. I am not sure that structural divergence is a good term for what they are analysing. The metrics used are not based on the RMSD of superposed relatives in an orthogroup and the mapping ratio (MR) is not often cited in the correlation analyses they perform. They typically cite CR, which is capturing proportion of identical residues in the mapped residues. For orthologs that retain the same function I would expect many residues to superpose well (as they typically find) so the CR ratio would reflect changes in physico-chemical properties to some extent. However, the CR ratio does not consider the degree of physico-chemical change e.g. they don't use any metrics to measure the degree of change in physicochemical properties. That could be interesting to consider in the future.

We thank the reviewer for this comment. We have both reworked the terminology and have conducted additional analyses. While the CR doesn't consider how much the residue's physicochemical properties change, comparing it to amino acid type conservation (which does consider these changes) shows a high correlation (Pearson r close to 1, (Response Figure R3 (left), included as Supplementary Figure S1.1D and lines 143-144 in the manuscript). This indicates that physicochemical properties are constraining conservation.

We also investigated the mean physicochemical properties of the entire protein, like isoelectric point, hydrogen-donor/-acceptor capabilities or hydrophobicity, but didn't see a strong relationship with the CR. However, if one compares the variability (standard deviation) of the amino acid properties with the CR, then one obtains stronger correlations. The strongest one is for the octanol-water partition coefficient (KOW), which is related to hydrophobicity (Response Figure R3 (right), included as Supplementary Figure S1.1E and lines 144-146 in the manuscript). We thus conclude that physicochemical properties are indeed a significant constraint captured by CR.

Response Figure R3, now Figure S1.1D,E: left: Conservation ratio versus type conservation ratio. Right: Variability of the octanol-water-coefficient versus conservation ratio

Furthermore, we have reworked the text to be semantically unambiguous (e.g. introduced the term 'sequence divergence within structurally mapped regions') and have added a definition of the terms used at the beginning of the results section (lines 161-163). We hope the terminology has improved. See also below (page 16 of this letter) where we highlight the updated text.

So the term is not going to be that different from sequence identity except for orthologs that don't superpose well. However, if I have understood correctly, for these remote orthologs that don't map well actual structural divergence is lost in the CR ratio as they normalize by the number of mapped residues rather than the length of the reference structure. I'm not sure why that method was chosen as it doesn't exploit the structural information from AlphaFold as fully as it might. The CR value they use

will miss some structural divergence e.g. inserted residues in the region of active site loops that may enable binding to a different substrate. The information on these unmappable inserted residues would be lost by considering CR on its own. Perhaps the authors can provide more explanation for this choice of metric.

While we acknowledge that focusing on CR may lead to the omission of information regarding larger structural changes, we came to the conclusion that this metric is best aligned with the specific needs and characteristics of our analysis, as large structural changes were rare within the orthogroups of the subphylum under consideration. In other words, MR values were consistently high while CR displayed greater variability. Therefore, CR provided a more informative metric for detecting variations where overall structure remained stable, but sequence divergence occurred.

Moreover, many of our follow-up analyses required residue-level comparisons. Techniques, such as analyzing structural features (e.g. active sites) based on reference structures and calculating dN/dS ratios, depend on a 1:1 mapping of residues across orthogroups. CR facilitates this level of detailed analysis, making it a good choice for our study. In summary, the rarity of broad structural changes within the orthogroups of the subphylum combined with the necessity for detailed residue-level analyses, resulted in us using Conservation Ratio over the Mapping Ratio. We have expanded the justification to the choice of our metric to the results section, lines 156-163.

Furthermore, for this reason, it might be good to use a term like 'divergence' rather than 'structural divergence' in the various studies or at least to discuss the caveats of the term they use, as they don't seem to be fully measuring structural divergence, unless I have misunderstood.

We apologise if the reviewer felt our choice of terminology was suboptimal. We have added a better definition of the terminology at the beginning of our manuscript (lines 161-163):

"During the course of the manuscript, we refer to sequence divergence in the structurally mapped region as 'divergence' (low CR) and sequence similarity in the structurally mapped region as 'conservation' (high CR)."

In addition, we simplified the terminology throughout the text, i.e. using the words "divergence" or "conservation", as suggested by the Reviewer, or circumscribe the result in other occasions.

4. How the orthogroups are generated is clear as they are generated using OrthoMCL and refined by structure comparison, but there seems to be subsequent clustering based on carbon source or species, which I found rather confusing. The authors describe these as subgroups of the orthogroups, which I assume means sets of orthogroups or are they subsets within orthogroups? It was a bit confusing and it

may be helpful to rephrase or have an example in supplementary to illustrate this better.

We apologize for the confusing wording surrounding the analysis we present for Figure 2A, which illustrates the only analysis in the manuscript that used clusters based on carbon source preference of the species. Therein, we split each orthogroup into two temporary subsets of species based on the growth of the species in 21 different media (Kurtzman et al., 2011; Lu et al., 2021) to test for the impact of metabolic niche preferences on enzyme structures. In other analyses, we cluster the structures based on the structurally refined OrthoMCL alignments ((Shen et al., 2018), Methods)).

To improve the wording, we now include a description of this subclustering in the methods section (lines 1102-1104), the caption of Figure 2 (line 168-169) and a Supplementary Figure S2.1A that illustrates which species are contained in each subgroup for each phenotype and shows the phylogenetic relationships between species.

5. The observation that enzymes involved in anaerobic fermentation are more constrained is very interesting but I was not sure I completely understood why this might be the case and would have appreciated more discussion. Is it because anaerobic fermentation is the more cost-effective process and residue changes could impact detrimentally on that?

We thank the reviewer for this point and have expanded the discussion about this point (lines 570-578). In brief, a shift from fermentation to oxidative metabolism is considered one of the major metabolic transitions in eukaryotic cells (Di Bartolomeo et al., 2020). It imposes differences in energetic output and efficiency as oxidative metabolism has a better stoichiometry for ATP production, but it's more costly in terms of resource allocation (Nilsson and Nielsen, 2016). The shift also imposes differences in carbon balance and transport processes and creates constraints on redox metabolism. In many cases, the changes from oxidative to fermentative metabolism are the dominant metabolic module in yeast and other microbes, and changes in metabolic pathways associated with fermentation are amongst the most dominant signals of domestication (Tengölics et al., 2024). In human cells, similar metabolic shifts have been associated with the Warburg effect in some cancer cells, or occur during immune cell activation (Stinccone et al., 2015). Hence, we speculated that these metabolic shifts could also impose a series of constraints on the enzyme evolution, which our analysis then confirmed. We have expanded the discussion accordingly (Supplementary Figure S.2.1B, Updated Figure 2A, see also response to Reviewer #1 comment to Figure 2a).

6. Some other observations are very easy to grasp such as observations that orthologous groups with more protein interactors or involved in large complexes were

more likely to be constrained. Furthermore, it is also intuitive that critical processes like central carbon metabolism and highly constrained enzymes like oxido-reductases have high CR. For enzymes like the hydrolases could the low CR reflect residue changes that are optimising the chemistry or modifying specificity to better exploit nutrient opportunities? I felt that tuning of the chemical reaction was hinted at in the section on changes in flux but it was a bit hard to follow that thread between sections.

We thank the reviewer for the positive feedback and for bringing up an interesting hypothesis, that the CR of hydrolases may reflect residue changes to better exploit nutrient opportunities. Although we cannot directly test the ecological aspects (there is very limited information about the nutrient opportunities present in the niches in which the examined yeast species evolved), we have examined the enzymatic aspects; i.e. the relationship of the overall CR and the binding sites, which reflects the evolution of substrate specificity. (I) When comparing the CR of the binding sites of hydrolases to the CR of all binding sites for both of our binding site definitions we observed no significant differences (Response Figure R4 (center, right), now included as Supplementary Figure S6.1G,H). (II) We analyzed the divergence of hydrolases in subsets of species as defined by nutrient usage (i.e. the analysis from Figure 2A for hydrolases instead of for all orthogroups). Also here, we did not notice phenotype groupings resulting in significant differences in CR between groups for hydrolases, indicating that, at least for these phenotypes, hydrolases are not more divergent or conserved than than other enzymes. (III) Also, hydrolases did not have abnormally low dN/dS values which indicates that they have typical levels of purifying selection.

However, for hydrolases, unlike in other enzymes, we detect no correlation between conservation rate and flux variability (Figure 3D). Furthermore, flux variability is significantly higher compared to all enzymes (Response Figure R4 (left, now included as Supplementary Figure S6.1C)). It is indeed possible that this variability in flux could represent an optimization of chemistry to better support a specialized metabolism. In order to improve this section of the manuscript, we have included a specific discussion about the specific properties of hydrolases (Lines 616-622, See also Reviewer #1 question on Figure 2C)

Response Figure R4, now Figure S6.1C,G,H: Mean flux variability (left) and binding site conservation for center: the CSb and right: the UPb binding site definitions for different enzyme classes. *** indicates a p-value < 1e-4, two-sided Wilcoxon-Mann-Whitney test.

7. Another observation that I found hard to intuitively grasp was that enzymes with highly conserved surfaces tend to be less costly. Could the authors suggest some reasons for this? Could this be because these enzymes are involved in complexes with high abundance, and it is the abundance that relates to the low cost? Perhaps I missed some details but there are many aspects to hold in the mind, which makes it hard to follow sometimes.

We thank the reviewer for this comment. The finding of a relationship of less costly surfaces on the most conserved enzyme structures, is consistent with our observation that evolution towards cost optimization is most prevalent for the most abundant enzymes, while inner structural elements and the substrate binding sites are more structurally constrained and therefore less prone to cost optimization compared to the surface. We have added this explanation to lines 387-390 and the discussion (lines 596-602).

8. It is interesting that the authors find that a significant proportion of the conserved residue clusters do not overlap with binding sites. It would be good to have a bit more clarity on this. Not all proteins have direct experimental information on binding sites. It would be helpful to know how many of the proteins and orthogroups in their dataset did have experimental functional sites data extracted from PDB structures.

It is also possible that some of the ligands in PDB structures are not cognate ligands - so some cognate binding residues could be missing. These clusters that don't overlap known binding sites could also reflect residues involved in allostery, which are not likely to be comprehensively annotated in PDB or UniProt. The authors may want to discuss these issues.

We thank you for the comment. We have expanded this section of the manuscript, to highlight better that binding sites are only one reason for conserved residue clusters on the surface. Another main reason is the presence of protein-protein coordination sites (see the additional analyses that we conducted as part of the reply to Reviewer #1 comment to Figure 5E, pages 8-10), and we agree with the reviewer that further reasons could be not-yet-annotated ligand binding sites, non-cognate ligands, allosteric effectors, as well as other structural interactions, such as stabilizing interactions within the protein core are equally plausible causes too (lines 507-508, 610-613). Our analysis will miss such elements if they are currently unknown, and we have added a relevant caveat to the manuscript.

For the second question, all our binding site annotations are in fact extracted from experimental data. Binding site information from existing PDB structures for 140 of the reference proteins, and 317 binding sites were annotated in Uniprot. We have now better highlighted this information (the caption of Figure 5G (line 428) and Figure S5.1F (line 1599), Supplementary Table S5.1).

Furthermore, I wasn't sure whether the authors were suggesting that the clusters

were missing some of the characterised binding sites, which would be a rather surprising observation. Perhaps that section could be phrased a bit more clearly.

It is correct that a subset of the annotated binding sites does not overlap with a cluster of highly conserved residues. This is because some binding sites are rather small (e.g. metal ion binding sites), others are not well annotated, while others are part of complexes and not captured by the monomeric structures. We have adjusted the text accordingly (lines 509-511)

9. As regards residue preferences, could the correlation regarding glycine content and conservation in random coil reflect the fact that glycine is often used for turns in the protein fold, which would tend to occur in residues at the surface of the protein not allocated to secondary structures? Do the authors see any trends for conserved glycine residues being associated with tight turns for example?

We thank the reviewer for this helpful comment. To analyze the glycine content in tight turns, we compared conservation ratios for residues from turns of different length. Overall, tight turns (< 7 amino acids) are more conserved than random coil structures (Response Figure R5 (top,left), now Supplementary Figure S5.2B). The relative glycine content of tight turns was higher than for all mapped random coils, with shorter turns showing a higher glycine content, most probably due to their unique backbone dynamics (Response Figure R5 (top,right), now Supplementary Figure S5.2E). As a negative control, alanine, which we expect to be less present (since it is known for its helix formation properties), decreases in tight turns compared to all mapped random coil regions (Response Figure R5 (bottom,left), now Supplementary Figure S5.2F). Furthermore, glycine shows a higher conservation in tight turns compared to glycine conservation in the overall protein (Response Figure R5 (bottom,right), now Supplementary Figure S5.2G). We added a subsection to the Supplementary Note 5 (lines 826-832) and one sentence to the results (lines 443-444) to highlight these observations.

Response Figure R5, now Figure S5.2B,E,F,G: (top, left) Conservation for different turn length and all random coil regions. (top right) Relative glycine or (bottom, left) alanine content for different turn sizes and the random coil region. (bottom, right) Glycine conservation in different structural elements. *** indicates a p-value < 1e-4, two-sided Wilcoxon signed-rank test.

In addition, as the authors point out they are unlikely to have surveyed random coil portions comprehensively as these are quite likely to be unmapped for many relatives in an orthogroup. The residues that map are indeed perhaps more likely to be involved in conserved regions valuable for the conformation.

We thank the reviewer for this comment. The mapped residues of random coils are not more conserved on average than other mapped residues (Response Figure R6 and new Supplementary Figure S5.2A). On the other hand, however, they are also not less conserved than other secondary structural elements (Supplementary Figure S5.1D). Taken together this indicates that the mapped random coil residues we surveyed are just as important for secondary structure conformation as helical or extended regions. One particularly important subset of mapped random coil residues,

however, are tight turns, which are more conserved than the overall protein. We added this information to the Supplementary Note 5 (lines 813-816).

Response Figure R6, now Figure S5.2A: Conservation ratio for different structural elements. *** indicates a p-value < 1e-4, two-sided Wilcoxon signed-rank test.

10. The observation of sites under positive selection locating at protein interfaces in structurally divergent orthogroups is another interesting observation and a phenomenon that seems reasonable given the fitness benefits from the enhanced stability of the oligomers. I was quite surprised there was not a signal for other types of functional sites though. Perhaps this suggests that most orthogroups are in fact associated with functionally identical relatives.

We apologise if our choice example (Pox1p), presented in the previous version of our manuscript, may have led to the impression that we observe positive selection only in protein-protein interaction (PPI) sites. As discussed above in the reply to Reviewer #1, we have expanded this analysis and have identified residues of positive selection in different structural elements in different enzymes (Lines 471-478, Supplementary Note 6.2). Please note that as part of our efforts to shorten the manuscript and make it more accessible, we have shortened this part in the main text considerably. An expanded Supplementary Figure S5.4 now demonstrates different structural elements where residues are positively selected in PPI sites, surface, and next to the catalytic core.

11. Since the orthogroups will belong to larger evolutionary superfamilies having particular structural characteristics, I was curious to know whether the specific structural arrangements that these superfamilies adopt in their core could also aid or constrain structural divergence. It may be interesting to map the orthogroups to superfamilies in ECOD or SCOP and determine whether some superfamilies are

more enriched in structurally divergent orthogroups. However, the article is already very rich in detail and that might be something for a future study.

We thank the reviewer for this suggestion. Unfortunately, in our dataset, we did not have enough orthogroups within a given superfamily in order to conduct a meaningful analysis. We were, however, able to analyse higher domain structural levels from the CATH framework (i.e. the architecture and fold levels) by extracting structural domains for each of our orthogroups from The Encyclopedia of Domains (TED) database (Lau et al., 2024). For this, we obtained domain data for any protein in our dataset which was also present in the AlphaFoldDB.

We performed an enrichment analysis on CATH ids for highly conserved as well as highly divergent orthogroups. We tested enrichment on different hierarchy levels like architecture or folds but only detected minor enrichment for most selections. One reason for this might be the low number of occurrences (sample sizes <10) since AUCs close to 0.8 were achieved, but with no significant (adjusted) p-values. For the most divergent orthogroups the closest hit was for architecture 1.20, the Up-Down-Bundle topology (adj. p-value < 0.1, AUC = 0.64). For the most conserved orthogroups, the only significant CATH architecture was 1.10, the orthogonal bundle (adj. p < 1e-3, AUC = 0.67, n = 55 reference structures). We added the respective analysis to the main text (lines 457-458).

At the topology level of the CATH hierarchy (the 3rd level, also called the 'fold' level), there was no significant enrichment for the most divergent orthogroups based on our adjusted p-value threshold, however, the two most common domains were slightly enriched. The Rossman fold (3.40.50) had the lowest unadjusted p-value of any of the domain topologies (adjusted p-value < 0.05, AUC = 0.55, n = 184 reference structures) and the TIM barrel (3.20.20, n = 58 reference structures) had an unadjusted p-value < 0.1 and an AUC = 0.60.

We added this analysis to the manuscript (lines 450-459). Also, please see related details on domain analysis in our response to Reviewer #3, comment #3 on page 26.

I found this article extremely interesting but it is incredibly dense. I read it multiple times but still wasn't sure if I fully appreciated the meaning of all the findings. For example, to what extent does diversity reflect specific benefits? For example, do residue changes alter enzyme specificity in order to adapt to different environments? Or do residue changes tune the chemistry and the K_{cat}/flux, or do residue changes optimize protein-protein interactions, thereby stabilizing complexes? I think the authors are suggesting that residue changes could be contributing to optimization of multiple properties, and the balance depends on the particular enzyme and pathway involved. However, the study covers so much that at times it is quite hard to keep track of the key fundamentals amongst the complexity. Nevertheless, I found it exciting and deeply interesting.

We thank the reviewer for all the input they have provided. In their concluding comment, they refer to some of the higher-level questions, like the ecological forces that act on metabolism such as niche optimisation, or the impact the structural adaptations have on the stability of macromolecular complexes. We have reworked the introduction and discussion to put a better emphasis on the higher-level questions. We have also worked on the presentation of our manuscript, to make it simpler and shorter, and hope that the new version is more accessible.

Referee #3:

This study explores how metabolism and structural evolution influence enzyme evolution over 400 million years. The findings reveal that:

1. Overall Metabolism: Metabolic processes shape enzyme structures, with evolution linked to reaction mechanisms and metabolic flux variability.
2. Metabolic Network Organization: The structure of the metabolic network influences enzyme evolution, affecting how enzymes interact within the network.
3. Enzymes' Molecular Properties: Evolutionary optimization varies across enzyme domains, prioritizing small-molecule binding sites, with surface residues adapting for protein-protein interactions.

Overall, I find the paper highly original and significant, with a clear, well-written manuscript that effectively links different disciplines. The findings are novel and offer valuable insights into how evolutionary constraints impact protein structure within a cellular (metabolic) environment. The study is robust in its data and conclusions, and it holds great relevance for protein scientists, molecular evolutionists, and the microbiology/synthetic biology community.

However, I have three major concerns/comments:

1. In the paragraph starting at line 259 “in parallel, we perform Go-slim...” the authors concluded that the metabolic specialization drives the structural evolution of enzymes that belong to these metabolic pathways. However, they never discuss the null hypothesis: what about if it is the other way around? What about if the enzyme structure shapes the metabolism? In principle first we have the emergence of enzymes and then they organise themselves in metabolic pathways.

We thank the reviewer for their input. We have expanded the discussion section to incorporate this alternative hypothesis. In brief, if the nature of enzymatic reaction catalysed would not be the main constraint for structure diversity, we would expect divergence of the chemical reactions within the orthogroups, and we would also expect a reverse hierarchy in structural evolution; e.g. we would expect more structural evolution in the binding sites relative to other structural features, both to

drive divergence of biochemical reactions, but also towards cost optimisation. Based on the data that is available, we see little evidence for such changes. For example, as aforementioned, we could not trace cases where an EC number of 3rd order and higher changed within an orthogroup (see reply to comment #1 of Reviewer #2 on pages 13-14). Moreover, the binding sites are the only structural elements that escape cost optimization. Certainly, there is the caveat that experimental binding site annotations and enzymological characterizations are not complete, but on the other hand, our conclusions are based on all annotations for yeast metabolic enzymes within the whole compendium of reference structures deposited in both PDB and Uniprot. It is thus fair to say that substantial changes in the nature of chemical reactions catalyzed within orthogroups at the level of evolution that our study covers, are rare. However, our data certainly does not exclude that in some cases there could have been larger shifts, e.g. across different kingdoms, that could have been caused, at least in part, by larger structural rearrangements.

Furthermore, our data is consistent with the hypothesis that structural changes in enzymes could allow switches to similar reactions, which are believed to be important for the evolution of new metabolic pathways, for instance if increased formation of a promiscuous product becomes a selectable trait. In our dataset, we have some evidence for possible changes between similar reactions in five orthogroups (see Reviewer #2, comment #1). We highlight one of these as an example, the orthogroup containing the related enzymes *Str2p* and *Hsu1p* in *S. cerevisiae*. *Hsu1p* allows inorganic sulfur fixation under specific conditions while *Str2p* does not (Yu et al., 2022; Sonal et al., 2023; Oss et al., 2022). We agree that these additions to the manuscript increase its value, and have adapted the discussion accordingly (lines 514-545, 634-639, Supplementary Note 3.2).

2. My second concern follows the previous comment: if any of the species has any specialised pathways not common to others, is it possible to explore these ones? And detect the differences? This data would add robustness to their results and discussion.

While most metabolic pathways are shared between the species in the subphylum, there indeed are a few metabolic capacities which are specific to some species. In order to address the Reviewer's comment, we focussed on the presence of a functional xylose utilisation pathway (Kwak and Jin, 2017). Of the 26 species 12 can grow on xylose, 8 could not, and 6 had a variable phenotype (new Supplementary Figure S2.1A). We compared the structure of enzymes that are associated with xylose utilization between the species which can grow on D-xylose and those that cannot. We found that splitting the species in this way led to the most significant differences in CR between groups (with respect to reference proteins from *S. cerevisiae*, new Figure 2B). Bearing in mind that species with similar phenotypes group together phylogenetically (see response to Reviewer #1, question to Figure 2a), we can gain additional insight by studying the pathways related to xylose utilization in a rank-based manner, and by examining genes and pathways that have

bigger differences in CR between species-defined subgroups than average. We now detail connections between xylose utilization and differences in specific enzymes involved in processing xylose in the text and find that several enzymes related to xylose utilisation have differences in conservation between xylose utilising and non-utilising species. This data thus strengthens our conclusion that the metabolic specialisation affects enzyme structural evolution (Lines 209-219, new Figure 2B, new Supplementary Figure S2.1, and Supplementary Note 3.1 to put our findings for xylose utilization in context).

3. To strengthen the manuscript, I suggest incorporating an analysis of fold usage, given the structural analysis already performed. It would be evolutionarily interesting to see if there is a bias in fold usage and whether the most catalytically diverse folds, such as the TIM barrel and Rossmann fold, appear in this context. Including this analysis would add another layer of depth to the study.

We thank the reviewer for this suggestion which aligns with the suggestion from Reviewer #2 and which we agree would add a deeper level of understanding to our analysis. As described in our response to Reviewer #2, comment #11 on page 23, we have included an analysis of domain presence in our revised manuscript. Moreover, we show that amino acids in a domain are slightly more conserved than amino acids outside of domains (two-sided Wilcoxon signed-rank test). The two most dominant folds in our dataset (> 30 entries), the Rossmann fold (CATH ID 3.40.50, n = 184 reference structures) as well as the TIM-barrel (CATH ID 3.20.20, n = 58 reference structures) are no exception. While they were not enriched in the most divergent orthogroups, on the basis of our adjusted p-value threshold they were more conserved than the overall protein on average, with a remarkable effect size for the TIM-Barrel (All domains: p-value < 1E-44, adj. Cliff's Delta = 0.64; CATH ID 3.40.50 (Rossmann-Fold): n = 184, p-value < 1E-4, adj. Cliff's Delta = 0.36; CATH ID 3.20.20 (TIM-Barrel): n = 58, p-value < 1E-7, adj. Cliff's Delta = 0.86). We added this information to the main text (lines 450-458) and included the Response Figure R7 as Supplementary Figure S5.1E.

Response Figure R7, now Figure S5.1E: Domain conservation ratio versus overall conservation ratio. The two most dominant folds the Rossmann-Fold (purple) and the TIM-Barrel (yellow) are highlighted.

Other minor comments/corrections:

Abstract line 47: “on the variability of flux (?) rather than...”

We reworked the abstract and took care to adjust the wording carefully.

Intro line 62: “enzyme plasticity is important for across disciplines”

We reworked the introduction to better emphasize the higher-level questions. As a consequence, the sentence was replaced.

Results line 173: The authors mention 7 major enzyme classes, the last one is “2 translocases”. These translocases are not addressed anywhere else in the paper; is there a reason? If so, can it be added in the text? If not, can this class be discussed as well (e.g. Fig. 3c includes only the first six classes of enzymes)?

We thank the reviewer for the comment. We had to exclude the translocases from several contrasts for statistical reasons; as there are only 2 translocases, the sample size is too small for meaningful comparisons involving enzyme conservation and many metabolic properties (e.g. Figure 3C and 3D). These orthogroups were, however, included in our overall analysis. We improved the statement to clarify this (Lines 248-250).

Line 423, “We calculated the average cost per amino acid for each protein averaged over the orthogroup using different cost metrics”: Does this mean that the authors determine the average cost per amino acid for each yeast species? Is this consistent across all species?

We thank the reviewer for this comment. Reviewer #1 also asked for more detailed background information on the cost metrics and the degree to which they are species-specific. We included text in the manuscript (lines 360-363) and a Supplementary Note (Supplementary Note 4) that reviews the different cost metrics applied (see also comment to Figure 4 to Reviewer #1).

References

- Aulakh, S.K., Lemke, O., Szyrwił, L., Kamrad, S., Chen, Y., Hartl, J., Muelleder, M., Nielsen, J., Ralser, M., 2024. The molecular landscape of cellular metal ion biology. <https://doi.org/10.1101/2024.02.29.582718>
- Blank, L.M., Kuepfer, L., Sauer, U., 2005. Large-scale ¹³C-flux analysis reveals mechanistic principles of metabolic network robustness to null mutations in yeast. *Genome Biol.* 6, R49. <https://doi.org/10.1186/gb-2005-6-6-r49>
- Daran-Lapujade, P., Jansen, M.L.A., Daran, J.-M., van Gulik, W., de Winde, J.H., Pronk, J.T., 2004. Role of transcriptional regulation in controlling fluxes in central carbon metabolism of *Saccharomyces cerevisiae*. A chemostat culture study. *J. Biol. Chem.* 279, 9125–9138. <https://doi.org/10.1074/jbc.M309578200>
- Di Bartolomeo, F., Malina, C., Campbell, K., Mormino, M., Fuchs, J., Vorontsov, E., Gustafsson, C.M., Nielsen, J., 2020. Absolute yeast mitochondrial proteome quantification reveals trade-off between biosynthesis and energy generation during diauxic shift. *Proc. Natl. Acad. Sci.* 117, 7524–7535. <https://doi.org/10.1073/pnas.1918216117>
- Gombert, A.K., Moreira dos Santos, M., Christensen, B., Nielsen, J., 2001. Network identification and flux quantification in the central metabolism of *Saccharomyces cerevisiae* under different conditions of glucose repression. *J. Bacteriol.* 183, 1441–1451. <https://doi.org/10.1128/JB.183.4.1441-1451.2001>
- Jouhten, P., Rintala, E., Huuskonen, A., Tamminen, A., Toivari, M., Wiebe, M., Ruohonen, L., Penttilä, M., Maaheimo, H., 2008. Oxygen dependence of metabolic fluxes and energy generation of *Saccharomyces cerevisiae* CEN.PK113-1A. *BMC Syst. Biol.* 2, 60. <https://doi.org/10.1186/1752-0509-2-60>
- Kajihata, S., Matsuda, F., Yoshimi, M., Hayakawa, K., Furusawa, C., Kanda, A., Shimizu, H., 2015. ¹³C-based metabolic flux analysis of *Saccharomyces cerevisiae* with a reduced Crabtree effect. *J. Biosci. Bioeng.* 120, 140–144. <https://doi.org/10.1016/j.jbiosc.2014.12.014>
- Kurtzman, C., Fell, J.W., Boekhout, T., 2011. *The Yeasts: A Taxonomic Study*. Elsevier.
- Kwak, S., Jin, Y.-S., 2017. Production of fuels and chemicals from xylose by engineered *Saccharomyces cerevisiae*: a review and perspective. *Microb. Cell Factories* 16, 82. <https://doi.org/10.1186/s12934-017-0694-9>
- Lau, A.M., Bordin, N., Kandathil, S.M., Sillitoe, I., Waman, V.P., Wells, J., Orengo, C.A., Jones, D.T., 2024. Exploring structural diversity across the protein universe with The Encyclopedia of Domains. *Science* 386, eadq4946. <https://doi.org/10.1126/science.adq4946>
- Lu, H., Li, F., Yuan, L., Domenzain, I., Yu, R., Wang, H., Li, G., Chen, Y., Ji, B., Kerkhoven, E.J., Nielsen, J., 2021. Yeast metabolic innovations emerged via expanded metabolic network and gene positive selection. *Mol. Syst. Biol.* 17, e10427. <https://doi.org/10.15252/msb.202110427>
- Nilsson, A., Nielsen, J., 2016. Metabolic Trade-offs in Yeast are Caused by F1F0-ATP synthase. *Sci. Rep.* 6, 22264. <https://doi.org/10.1038/srep22264>
- Opulente, D.A., Rollinson, E.J., Bernick-Roehr, C., Hulfachor, A.B., Rokas, A., Kurtzman, C.P., Hittinger, C.T., 2018. Factors driving metabolic diversity in the budding yeast subphylum. *BMC Biol.* 16, 26. <https://doi.org/10.1186/s12915-018-0498-3>
- Oss, S.B.V., Parikh, S.B., Coelho, N.C., Wacholder, A., Belashov, I., Michaca, M.,

- Xu, J., Kang, Y.P., McCourt, K.M., McKee, J., Ideker, T., VanDemark, A.P., DeNicola, G.M., Carvunis, A.-R., 2022. Unexpected growth of a classic yeast auxotroph. <https://doi.org/10.1101/2022.01.19.476918>
- Raghevedran, V., Gombert, A.K., Christensen, B., Kötter, P., Nielsen, J., 2004. Phenotypic characterization of glucose repression mutants of *Saccharomyces cerevisiae* using experiments with ¹³C-labelled glucose. *Yeast* Chichester Engl. 21, 769–779. <https://doi.org/10.1002/yea.1136>
- Shen, X.-X., Opulente, D.A., Kominek, J., Zhou, X., Steenwyk, J.L., Buh, K.V., Haase, M.A.B., Wisecaver, J.H., Wang, M., Doering, D.T., Boudouris, J.T., Schneider, R.M., Langdon, Q.K., Ohkuma, M., Endoh, R., Takashima, M., Manabe, R., Čadež, N., Libkind, D., Rosa, C.A., DeVirgilio, J., Hulfachor, A.B., Groenewald, M., Kurtzman, C.P., Hittinger, C.T., Rokas, A., 2018. Tempo and Mode of Genome Evolution in the Budding Yeast Subphylum. *Cell* 175, 1533–1545.e20. <https://doi.org/10.1016/j.cell.2018.10.023>
- Sonal, Yuan, A.E., Yang, X., Shou, W., 2023. When is an auxotroph not an auxotroph: how budding yeast lacking MET17 collectively overcome their metabolic defect. <https://doi.org/10.1101/2023.05.18.541364>
- Stincone, A., Prigione, A., Cramer, T., Wamelink, M.M.C., Campbell, K., Cheung, E., Olin-Sandoval, V., Grüning, N.-M., Krüger, A., Alam, M.T., Keller, M.A., Breitenbach, M., Brindle, K.M., Rabinowitz, J.D., Ralser, M., 2015. The return of metabolism: biochemistry and physiology of the pentose phosphate pathway. *Biol. Rev. Camb. Philos. Soc.* 90, 927–963. <https://doi.org/10.1111/brv.12140>
- Tengölics, R., Szappanos, B., Mülleder, M., Kalapis, D., Grézal, G., Sajben, C., Agostini, F., Mokochinski, J.B., Bálint, B., Nagy, L.G., Ralser, M., Papp, B., 2024. The metabolic domestication syndrome of budding yeast. *Proc. Natl. Acad. Sci.* 121, e2313354121. <https://doi.org/10.1073/pnas.2313354121>
- The UniProt Consortium, 2025. UniProt: the Universal Protein Knowledgebase in 2025. *Nucleic Acids Res.* 53, D609–D617. <https://doi.org/10.1093/nar/gkae1010>
- Yu, J.S.L., Heineike, B.M., Hartl, J., Correia-Melo, C., Aulakh, S.K., Lehmann, A., Lemke, O., Agostini, F., Lee, C.T., Demichev, V., Messner, C.B., Mülleder, M., Ralser, M., 2022. Inorganic sulfur fixation via a new homocysteine synthase allows yeast cells to cooperatively compensate for methionine auxotrophy. <https://doi.org/10.1101/2022.03.14.484209>

Referees' comments

Referee #1:

The authors responded reasonably to my concerns from the first round of review. They performed several additional requested computational analyses. My one major remaining concern, which I acknowledge is a challenge to address, relates to the lack of experimental validation of a meaningful (but not onerous) subset of computational predictions. There was essentially none of this performed.

We appreciate the reviewer's suggestion regarding the inclusion of additional experimental data. While our original manuscript relied primarily on AlphaFold2-predicted protein structures, the revised version incorporates a substantial number ($n = 436$) of experimentally determined protein structures from the Protein Data Bank (PDB). In fact, we use all available enzyme structures that match our study's targets to validate and support our findings.

Beyond protein structures, we have integrated a wide range of experimentally derived datasets. We list those in Supplementary table 7.1, including metal-binding data, enzyme commission (EC) annotations, proteomics data, protein complex and protein-protein and protein-substrate interaction data, as well as growth phenotypes. Specifically in response to the reviewer's comments, we also added experimentally determined metabolic fluxes (Figure ED3.1c), protein-protein coordination sites (Figures ED5.1i and ED5.2c), and data relevant to xylose utilization (Supplementary Note 3.1) as part of the revision.

It is correct that—aside from the proteomics data—we did not generate these experimental datasets ourselves. However, using comprehensive, community-curated datasets was a deliberate and optimal strategy for our study. These high-quality resources enabled us to perform large-scale, systemic validation that would not have been feasible with smaller, less systematic datasets. This approach not only strengthens the robustness of our findings but also aligns our work with widely used community resources, enhancing its relevance and reproducibility. We would also like to emphasize that the extensive integration of these experimental datasets positions our study among the most data-rich currently available in this field.

Additional points:

Regarding the statement in the response: "Furthermore, oxidoreductases are indeed more conserved overall than other enzymes, however, this is not solely due to their dependency on a cofactor: this relationship only holds true as long as one includes the oxidoreductases of central and high-flux pathways - glycolysis, gluconeogenesis and glutamate biosynthesis - that are rich in oxidoreductases. Once one excludes the oxidoreductases participating in these pathways, the oxidoreductases are not more conserved than other enzyme classes."

If this is true, why not split out central and non-central oxidoreductases to highlight this relationship in the text? If this is there, I missed it.

We had a similar intuition as the reviewer, and in fact this analysis was already part of our revision. We apologize that we had not highlighted it sufficiently (Figure ED2.2d). We revised the text (now lines 197+198) to highlight the analysis better.

- Ln 447-448: Is the appropriate figure being referenced here? I don't see data for the loop region prediction error related question.

We apologize for the misleading description. The word "loop" was replaced with "random coil" to make it more appropriate. We also shifted the paragraph to a better position, to make the relationship clearer. Moreover, we added a detailed explanation to Supplementary Note 2.

- Ln 493: Please define the ML model, what specifics?

Here we used a histogram-based gradient boosting classification tree on physico-chemical properties. We have expanded the results section with the specifics and apologize for the lack of detail in our revision.

- Several of the figures seem to be missing colormaps (e.g. Fig. 3c) or appropriate definitions (e.g. "1" vs "0" in Fig. S2.1a) and descriptions (e.g. groups being compared for enrichment in S2.2).

We have reviewed all figures and provided appropriate definitions, colormaps and descriptions in the figure's caption now.

Referee #2:

I have checked the authors responses and revisions to the paper. I'm grateful for the detailed answers and the additional research that the authors performed in response to my questions and suggestions. I find the revised manuscript much clearer.

It's helpful to know that most orthogroup relatives share the same EC4 number. Apologies if I missed that detail before. It's useful to have the information on levels of experimental annotation.

I thank the reviewers for their analysis of fold groups in the TED classification and found it interesting and maybe not surprising that the Rossman and TIM barrel fold groups were slightly

enriched in the most divergent orthogroups. These fold groups are likely to be enriched in enzyme superfamilies which may have optimised the mechanism or efficiency of their reactions.

One minor comment remains which does not need to be implemented. It was helpful to have more explanation for the choice of TMscore. It might be helpful to remind readers of the equation for this score as that would make it clearer that it better captures global similarity than RMSD.

We thank the reviewer for highlighting this point. We have added more detail to our description in the methods about our choice of TM score to explain the score in a more nuanced way to the reader pointing out (1) that TM-score better captures global similarity by weighing closer distances between aligned atoms more heavily than larger distances, thereby preventing a single poorly aligned region from overly influencing the overall score, and (2) that TM-score contains a normalizing factor that makes it independent of the overall protein length for random proteins (i.e. close to 0.17), unlike RMSD, for which two random large proteins would tend to have a worse RMSD than two random small proteins. As suggested, we have also included the original equations on page 16 in the methods section.

I thank the authors again for a fascinating and rigorous article.

Referee #2 (Remarks on code availability):

I have not been able to review the code.

Referee #3:

The authors have made substantial improvements to the manuscript by thoughtfully addressing the concerns raised in my previous review. The additional explanations and clarifications provided significantly enhance the context for their observations and strengthen the overall observation.

We thank the author for their time and effort reviewing our manuscript and for their previous suggestions which have improved the manuscript.